# Neural Symbolic Regression of Complex Network Dynamics

## Abstract

Complex networks describe important structures in nature and society, composed of nodes and the edges that connect them. The evolution of these networks is typically described by dynamics, which are labor-intensive and require expert knowledge to derive. However, because the complex network involves noisy observations from multiple trajectories of nodes, existing symbolic regression methods are either not applicable or ineffective on its dynamics. In this paper, we propose Physically Inspired Neural Dynamics Symbolic Regression (PI-NDSR), a method based on neural networks and genetic programming to automatically learn the symbolic expression of dynamics. Our method consists of two key components: a Physically Inspired Neural Dynamics (PIND) to augment and denoise trajectories through observed trajectory interpolation; and a coordinated genetic search algorithm to derive symbolic expressions. This algorithm leverages references of node dynamics and edge dynamics from neural dynamics to avoid overfitted expressions in symbolic space. We evaluate our method on synthetic datasets generated by various dynamics and real datasets on disease spreading. The results demonstrate that PI-NDSR outperforms the existing method in terms of both recovery probability and error.

## 1 Introduction

Complex networks (Gerstner et al., 2014; Gao et al., 2016; Bashan et al., 2016; Newman et al., 2011) describe important structures in nature and society, which is composed of a set of nodes and a set of edges that connect them. Complex networks can model various real-world systems, including social networks (Kitsak et al., 2010), epidemic networks (Pastor-Satorras & Vespignani, 2001), brain networks (Laurence et al., 2019; Wilson & Cowan, 1972), and transportation networks (Kaluza et al., 2010). Extensive works (Zang & Wang, 2020; Murphy et al., 2021; Gao & Yan, 2022) have been conducted to analyze the dynamics of complex networks (Pastor-Satorras et al., 2015; MacArthur, 1970; Kuramoto & Kuramoto, 1984), which is crucial for understanding the underlying mechanisms of complex systems and predicting their future behaviors. Among them, the symbolic complex network dynamics is of great importance as it offers a clear and concise depiction of the internal mechanisms and their impact on the overall system behavior.

Obtaining the symbolic expressions for complex network dynamics is challenging and is usually done by mathematicians or physicists. The process of symbolic regression of complex network dynamics typically requires repetitive trial-and-error attempts and profound expertise in the relevant field. Therefore, it is often unlikely to find the symbolic expressions without any prior knowledge of the system (Virgolin & Pissis, 2022). The goal of this paper is to develop a machine-learning method to automatically learn the symbolic expressions of complex network dynamics from data.

Symbolic regression aims to discover the underlying symbolic formula from data (Schmidt & Lipson, 2009; Petersen et al., 2019; Cranmer et al., 2020; Shi et al., 2023). Various techniques have been applied to symbolic regression for dynamics, including genetic programming (Gaucel et al., 2014; Kronberger et al., 2020), sparse regression (Brunton et al., 2016), deep-learning-based genetic programming (Qian et al., 2022), and Transformer (d'Ascoli et al., 2024). However, all these methods are designed for symbolic regression of dynamics for a single trajectory. Directly applying them to complex network dynamics ignores common patterns between trajectories and leads to inefficient learning and poor performance (Gao & Yan, 2022), as the complex network is an evolving system

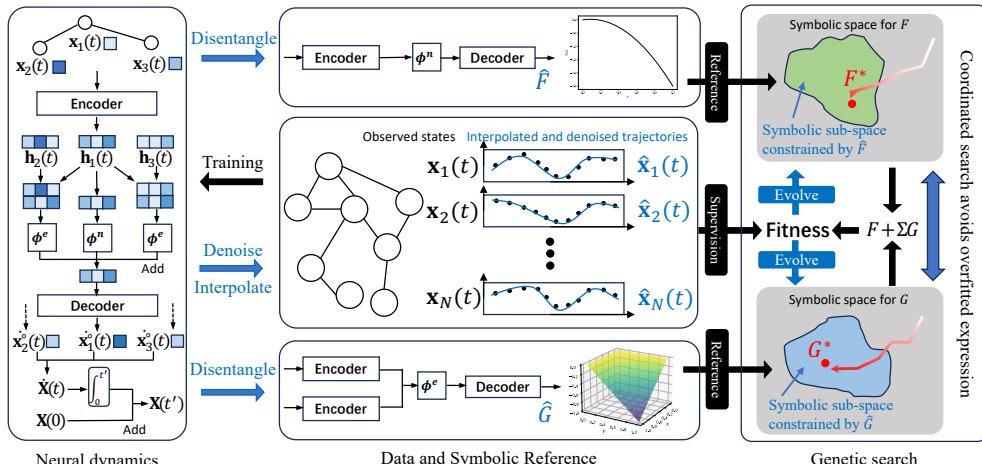

Figure 1: Our method is designed for symbolic regression of complex network dynamics. The proposed method interpolates and denoises complex network observations with neural dynamics to avoid inaccurate estimation of time derivatives and applies a coordinated genetic search algorithm to derive the symbolic expressions of complex network dynamics.

with multiple observed trajectories of nodes sharing the same underlying dynamics. To deal with multiple trajectories, recent work (Gao & Yan, 2022) proposes a Two-Phase Sparse Identification of Nonlinear Dynamics (TP-SINDy) for symbolic regression of complex network dynamics. The two-phase optimization in TP-SINDy effectively narrows down the symbolic search space and improves its performance. However, TP-SINDy relies on the estimated time derivatives of the node states, which are noisy and inaccurate. Additionally, since it is built on SINDy (Brunton et al., 2016), there exists a pre-defined function library for regression. This can make it challenging to recover the symbolic expression beyond the confines of the function library. In this paper, we propose Physically Inspired Neural Dynamics Symbolic Regression (PI-NDSR) for the complex network dynamics based on the observed trajectory, which is more accurate and robust than the previous methods.

The proposed method (Fig. 1) consists of two parts: (a) A physically inspired neural dynamics (neural dynamics refers to the dynamics represented by neural network) disentangles the node dynamics and edge dynamics to augment and denoise trajectories with the observed trajectory interpolation. Neural dynamics provides high-quality supervision signals for calculating fitness in the genetic search algorithm and gives references for symbolic expressions to enhance search efficiency. (b) A coordinated genetic search algorithm regresses the symbolic expression of network dynamics that uses references of node dynamics and edge dynamics from neural dynamics to constrain the symbolic search space, which helps avoid overfitted expressions. Compared with the previous methods which relies on the estimated time derivative, PI-NDSR applies to more general complex network observations with larger noise and sampling intervals and does not rely on a pre-defined function library. The proposed method is evaluated on synthetic datasets generated with various dynamics and a real dataset from disease spreading. The results demonstrate that our method surpasses the previous approach in terms of recovery probability and error. Our contributions are summarized as follows: (1) Using both the interpolated denoised trajectories and references of node/edge dynamics from physically-inspired neural dynamics for supervision, PI-NDSR proposes a coordinated genetic search algorithm to avoid overfitted expressions and improves the quality of symbolic expressions. (2) Compared to previous methods, PI-NDSR has better recovery probability and smaller error in the recovered symbolic dynamics.

**Notations** Matrix, vector, and scalar are denoted as bold capital letters $\mathbf{X}$, bold lowercase letters $\mathbf{x}$, and lowercase letters $x$, respectively. The element in $i$-th row and $j$-th column of matrix $\mathbf{X}$ is denoted as $\mathbf{X}_{ij}$. The $v$-th row of matrix $\mathbf{X}$ is denoted as $\mathbf{x}_v$. A complex network is denoted as $\mathbf{G} = (\mathsf{G}, \mathbf{X}(t), t \in \mathcal{T})$. $\mathsf{G} = (V, E)$ is the structure of complex network where $V$ is the set of nodes and $E$ is the set of edges. $\mathbf{X}(t) = [\mathbf{x}_1(t)^\top, \cdots, \mathbf{x}_N(t)^\top]^\top \in \mathbb{R}^{N \times d}$ is a $d$-dimensional node states of $N$ nodes at the timestamp $t$, and $\mathcal{T} = \{t_0, t_1, \cdots, t_{K-1}\}$ is the set of $K$ timestamps of complex network observations. $\dot{\mathbf{x}}(t)$ represents the time derivatives of $\mathbf{x}(t)$.

## 2 RELATED WORK

### 2.1 SYMBOLIC REGRESSION

Throughout the history of physics, extracting elegant symbolic expressions from extensive experimental data has been a fundamental approach to uncovering new formulas and validating hypotheses. Symbolic Regression (SR) is a notable topic in this context (Schmidt & Lipson, 2009; Petersen et al., 2019; Cranmer et al., 2020; Kamienny et al., 2022), aiming to mimic the process of deriving an explicit symbolic model that accurately maps input $X$ to output $y$ while ensuring the model remains concise. Traditional methods for deriving formulas from data have predominantly relied on genetic programming (GP) (Schmidt & Lipson, 2009; Koza, 1994; Worm & Chiu, 2013), a technique inspired by biological evolution that iteratively evolves populations of candidate solutions to discover the most effective mathematical representations.

More recently, due to the remarkable accomplishments of neural networks across diverse domains, there has been an increasing interest in leveraging deep learning for symbolic regression. Specifically, some recent works (Cranmer et al., 2020; Chen et al., 2021; Qian et al., 2022; Udrescu & Tegmark, 2020; Martius & Lampert, 2016; Mundhenk et al., 2021; Shi et al., 2023) have explored guiding genetic programming with the output of neural networks to improve the efficiency and accuracy of symbolic regression. This approach takes advantage of the powerful pattern recognition and generalization capabilities of neural networks to inform the evolutionary processes of genetic programming, resulting in more effective and efficient discovery of symbolic expressions. Another line of works (Kamienny et al., 2022; Biggio et al., 2021; Valipour et al., 2021; d'Ascoli et al., 2024) applies Transformer to symbolic regression and achieves comparable performance to GP-based methods.

### 2.2 COMPLEX NETWORK DYNAMICS LEARNING

With the development of machine learning in recent decades, many works have been proposed to learn the dynamics of complex networks from data. The work for learning dynamics of complex networks can be divided into two categories: dynamics learning with neural networks and dynamics learning with symbolic regression.

Dynamics learning with neural networks utilizes deep neural networks to learn dynamics (Hamrick et al., 2018; Zang & Wang, 2020; Liu et al., 2023). Neural dynamics usually follows the encode-process-decode paradigm (Hamrick et al., 2018; Zang & Wang, 2020) with an encoding neural network to pre-process the initial node states by mapping them to latent states. Then the latent states are processed by a graph neural network to capture the evolution and interaction of the complex network. Murphy et al. (2021) propose to use GNN to model the evolution of complex networks with regularly sampled observation. NDCN (Zang & Wang, 2020) models the continuity of the dynamics with graph neural ODE (Chen et al., 2018) and the interaction of the dynamics with GNN. Even though neural dynamics learning can capture the complex dynamics of complex networks, it is difficult to interpret the learned dynamics due to the black-box nature of neural networks. Recently, Liu et al. (2023) drop the encoder-process-decoder paradigm and achieve better performance on predicting long-term behavior of complex networks.

On the other hand, dynamics learning with symbolic regression aims to learn symbolic expressions of complex network dynamics from data. Two-phase sparse identification of nonlinear dynamical systems (TP-SINDy) (Gao & Yan, 2022) is proposed to obtain the dynamics of complex networks from data. The proposed method includes a function library including a wider range of elementary functions than the orthogonal basis functions in SINDy (Brunton et al., 2016) and a two-phase regression method to learn the dynamics of complex networks. However, TP-SINDy requires an accurate estimation of the time derivative of the node states and can only learn the symbolic expressions from the pre-defined function library. Our work uses the supervision signal and dynamics references provided by neural dynamics, based on genetic programming to learn the symbolic expressions of complex network dynamics.

## 3 PROBLEM SETUP

The complex network dynamics is defined by the following differential ordinary equation:

$$\dot{\mathbf{x}}_v(t) = F\left(\mathbf{x}_v(t)\right) + \sum_{u \in N_v} a_{vu} G\left(\mathbf{x}_v(t), \mathbf{x}_u(t)\right), \tag{1}$$

In (1), $\dot{\mathbf{x}}_v(t)$ denotes the time derivative of $\mathbf{x}_v(t)$, $F(\mathbf{x}_v(t))$ denotes the node dynamics term of node $v$, which includes processes like influx, degradation, or reproduction. $G(\mathbf{x}_v(t), \mathbf{x}_u(t))$ is the edge dynamics describing the interactions between node $v$ and node $u$, accounting for processes such as spreading and competition. $G$ is shared across all edges in the network because of the universality in network dynamics (Barzel & Barabási, 2013; Gao et al., 2016). $a_{vu}$ is the weight of the edge between node $v$ and node $u$, and $N_v$ is the set of neighbors of node $v$.

Given observations of the node states $\{\mathbf{X}(t) | t \in \mathcal{T}\}$, *the symbolic regression of complex network dynamics* (Gao & Yan, 2022) aims to find the symbolic expressions of function $F$ and $G$ in (1). Compared with traditional symbolic regression, we identify two main challenges of symbolic regression for network dynamics: (a) Directly regressing symbolic function on time derivative is inaccurate and unstable because the estimation of derivative $\dot{\mathbf{X}}(t)$ is noisy and its accuracy relies on the regular sampling intervals of the observation. (b) The network dynamics consist of two parts, namely node dynamics $F$ and edge dynamics $G$. In symbolic regression, it is possible that $F$ and $G$ jointly perform well on the observed (interpolation) trajectory but exhibit significant discrepancies in predicting node dynamics and edge dynamics compared to the real dynamics, which results in bad extrapolation performance (see examples in Appendix B.4). We refer to this phenomenon as the overfitting problem in symbolic space. In this paper, we propose a physically inspired symbolic regression algorithm, addressing the first challenge with interpolated network observations and the second challenge with a coordinated genetic optimization algorithm.

## 4 METHOD

To address the challenge of inaccurate estimation of $\dot{\mathbf{X}}(t)$ in symbolic regression, we instead utilize the state of the nodes as supervision. We train a neural network $f_\theta$ with parameter $\theta$ to denoise and interpolate the observed trajectory as the approximation for the true observation $\mathbf{X}(t)$. To address the challenges in symbolic regression of node dynamics and edge dynamics in (1), we design a coordinated genetic optimization algorithm based on the algorithmic alignment between neural network $f_\theta$ and network dynamics (1).

### 4.1 TRAJECTORY DENOISING AND AUGMENTATION WITH INTERPOLATED OBSERVATIONS

In symbolic regression, it is beneficial to regress symbolic expressions with many low-noise network observations. However, the network dynamics observation is usually sparse (i.e., sample time interval is large) and noisy. To address this issue, we propose Physically Inspired Neural Dynamics (PIND) to interpolate and denoise the network observations, obtaining denser and less noisy trajectories. Except for the interpolated trajectory, the neural dynamics aligns with network dynamics in (1) and provides an estimation of the node dynamics and edge dynamics, which is used as references for coordinated genetic optimization in deriving the symbolic $F$ and $G$.

Following existing works (Zang & Wang, 2020; Murphy et al., 2021), the neural dynamics is designed based on encode-process-decode paradigm (Battaglia et al., 2018). To obtain the denser and less noisy trajectory and the high-quality estimation for node dynamics and edge dynamics for genetic search, we incorporate the algorithmic inductive bias in (1) to design a physically-inspired neural dynamics and train the neural dynamics from the observed trajectory. In neural dynamics, the raw observation $\mathbf{x}_v(t)$ is first encoded by an encoding network Enc to obtain the latent state $\mathbf{h}_v(t)$, and the graph neural network is applied to calculate the time derivative of the latent state $\dot{\mathbf{h}}_v(t)$, which is later decoded by a decoding network Dec to obtain the time derivatives $\dot{\mathbf{x}}_v^\circ(t)$. Finally, differential ODESolver (Chen et al., 2018) is applied to $\dot{\mathbf{x}}_v^\circ(t)$ to evolve the neural dynamics to obtain more network observations.

Based on (1), the time derivative $\dot{\mathbf{x}}_v^\circ(t)$ for node $v$ is designed with inductive bias from (1):

$$\dot{\mathbf{x}}_v^\circ(t) = \mathsf{Dec}(\dot{\mathbf{h}}_v(t)), \quad \text{s.t.} \quad \begin{cases} \dot{\mathbf{h}}_v(t) = \phi^n\left(\mathbf{h}_v(t), t\right) + \sum_{u \in N_v} \phi^e(\mathbf{h}_v(t), \mathbf{h}_u(t), t), \\ \mathbf{h}_v(t) = \mathsf{Enc}(\mathbf{x}_v(t)), \mathbf{h}_u(t) = \mathsf{Enc}(\mathbf{x}_u(t)), u \in N_v, \end{cases} \tag{2}$$

where $\phi^n$ and $\phi^e$ are two MLPs aligning with the node dynamics and edge dynamics in (1), respectively, and $\mathbf{h}_v(t) \in \mathbb{R}^{d'}$ is the latent state of node $v$. In (2), the neural node dynamics $\phi^n$ captures the evolution of nodes influenced by their properties, and the neural edge dynamics $\phi^e$ captures the interactions between two end nodes of an edge. Therefore, PIND of node $v$ is written as

$$f_\theta(\mathsf{G}, \mathbf{X}(t_0), t)_v = \mathrm{ODESolver}(\dot{\mathbf{x}}_v^\circ(t), \mathbf{X}(t_0), t_0, t). \tag{3}$$

The alignment between neural dynamics (3) and dynamics formulation (1) enables better learning of complex network dynamics. To train the neural dynamics, we minimize the error between $f_\theta(\mathsf{G}, \mathbf{X}(t_0), t)_v, \forall v \in V$ and the observed trajectories $\{\mathbf{X}(t) | t \in \mathcal{T}\}$, i.e., $\min_\theta \sum_{v \in V, t \in \mathcal{T}} \|f_\theta(\mathsf{G}, \mathbf{X}(t_0), t)_v - x_v(t)\|_1$, with standard deep learning optimization techniques. After the training, we use the interpolated trajectory $\hat{\mathbf{X}}(t)$ from $f_\theta(\mathsf{G}, \mathbf{X}(t_0), t)_v, \forall v \in V$ as the supervision signal for symbolic regression and the estimated node dynamics and edge dynamics as references for the genetic search in deriving the symbolic $F$ and $G$ in Section 4.2.

## 4.2 COORDINATED GENETIC SEARCH FOR SYMBOLIC REGRESSION

To regress the symbolic formulas $F$ and $G$, we propose an algorithm called coordinated genetic search to obtain the symbolic expressions of complex network dynamics. Based on the physically-inspired design of (2), we can obtain the references for symbolic expressions by disentangling the neural dynamics into node dynamics and edge dynamics. Then, we use the neural references to coordinate the search process of symbolic expressions of $F$ and $G$ to avoid the overfitted expressions.

Because of the inductive bias and algorithmic alignment of (1), the neural dynamics often have better generalization performance (Xu et al., 2019; Veličković & Blundell, 2021). Therefore, the neural node dynamics and edge dynamics can be used as estimations of the node dynamics and edge dynamics in (1), which can be used as references for coordinating the search process of symbolic expressions of $F$ and $G$. The neural node dynamics and edge dynamics can be computed with the following equations:

$$\hat{F}(\mathbf{x}_v(t)) = \mathsf{Dec}\left(\phi^n(\mathsf{Enc}(\mathbf{x}_v(t)), t)\right), \hat{G}\left(\mathbf{x}_v(t), \mathbf{x}_u(t)\right) = \mathsf{Dec}\left(\phi^e(\mathsf{Enc}(\mathbf{x}_v(t)), \mathsf{Enc}(\mathbf{x}_u(t)), t)\right). \tag{4}$$

Unlike Cranmer et al. (2020), we do not conduct the genetic search on (4) directly because the neural dynamics are not accurate enough for the fitness calculation. Instead, coordinated genetic search uses the neural dynamics for reference and denoised interpolated trajectories for fitness calculation to guide the search process. In the coordinated genetic search, we have two populations containing symbolic node dynamics and edge dynamics, i.e., $\mathcal{F}$ and $\mathcal{G}$. However, in our problem setup, the populations $\mathcal{F}$ and $\mathcal{G}$ often have different distances to their corresponding ground-truth dynamics. Jointly evolving $\mathcal{F}$ and $\mathcal{G}$ may cause the population closer to ground truth to be overfitting and another population to be underfitting. Therefore, we select population to coordinate their evolution process by comparing the distances of $\mathcal{F}$ and $\mathcal{G}$ to the references $\hat{F}$ and $\hat{G}$ in (4). In the search process, we calculate the distances between populations $\mathcal{F}$ and $\mathcal{G}$ to the references $\hat{F}$ and $\hat{G}$, respectively with

$$d(\mathcal{F}) = \sum_{F \in \mathcal{F}} \|F - \hat{F}\|^2 \quad \text{and} \quad d(\mathcal{G}) = \sum_{G \in \mathcal{G}} \|G - \hat{G}\|^2, \tag{5}$$

where $\|\cdot\|$ represents the distance between functions and is the average of absolute error between two functions on randomly sampled points in our paper. Then, we select to evolve the population with a larger distance to the reference, i.e., if $d(\mathcal{F}) > d(\mathcal{G})$, we evolve the node dynamics population $\mathcal{F}$; otherwise, we evolve the edge dynamics population $\mathcal{G}$. The coordinated strategy in genetic search helps to avoid the overfitted expressions in symbolic space and improve the quality of the symbolic expressions of $F$ and $G$.

To evolve the selected population, we assess the fitness of each symbolic expression in the selected populations by comparing the error between the integral of combined symbolic node dynamics and edge dynamics with the interpolated trajectory. For the node dynamics population $\mathcal{F}$ and edge

dynamics population $\mathcal{G}$, the fitness of a symbolic node dynamics $F \in \mathcal{F}$ and a symbolic edge dynamics $G \in \mathcal{G}$ are respectively calculated with

$$\mathsf{f}_F = \mathrm{Mean} \circ \mathrm{BigK} \Big\{ \sum\nolimits_{v \in V, t \in T} -e \Big( \int_0^t F(\mathbf{x}_v) + \sum\nolimits_{u \in N_v} G(\mathbf{x}_v, \mathbf{x}_u) dt, f_\theta \left( \mathsf{G}, \mathbf{X}(t_0), t \right)_v \Big) \Big| G \in \mathcal{G} \Big\}, \quad (6)$$

$$\mathsf{f}_G = \mathrm{Mean} \circ \mathrm{BigK} \Big\{ \sum\nolimits_{v \in V, t \in T} -e \Big( \int_0^t F(\mathbf{x}_v) + \sum\nolimits_{u \in N_v} G(\mathbf{x}_v, \mathbf{x}_u) dt, f_\theta \left( \mathsf{G}, \mathbf{X}(t_0), t \right)_v \Big) \Big| F \in \mathcal{F} \Big\}, \quad (7)$$

where $e(\cdot, \cdot)$ is the error function between the symbolic integral value and interpolated trajectory, $T$ is the set of timestamps interpolated from training time ranges, BigK is the function that selects the $K$ largest negative errors from a set, Mean is the mean operator and $\circ$ is the composition operator. In (6) and (7), a large fitness indicates that the corresponding symbolic expression is close to the ground-truth dynamics because the integral of the symbolic expression of dynamics is close to the interpolated trajectory. Then, the expressions in the selected population are selected, crossed, and mutated by the genetic algorithm to generate the next population.

The coordinated genetic search algorithm is presented in Algorithm 1. In the algorithm, we first initialize the node dynamics and edge dynamics populations $\mathcal{F}^{(0)}$ and $\mathcal{G}^{(0)}$ with random symbolic expressions in line 1. In the evolution process, the populations $\mathcal{F}^{(i-1)}$ and $\mathcal{G}^{(i-1)}$ are evolved to generate the next populations $\mathcal{F}^{(i)}$ and $\mathcal{G}^{(i)}$ in lines 3-14. The process is repeated until the fitness from (6) and (7) is below a threshold (lines 6 & 11) or reaching the maximum number $M$ of iterations. In the end, the algorithm selects and outputs the symbolic node dynamics and edge dynamics with the lowest fitness in lines 16-18.

---

**Algorithm 1** Coordinated genetic search for symbolic regression

---

**Require:** Neural dynamics $f_\theta$, node dynamics reference $\hat{F}$, edge dynamics reference $\hat{G}$, $K$ for calculating fitness, maximum iteration $M$, threshold $\epsilon$.
1: Initialize the node dynamics population $\mathcal{F}^{(0)}$ and edge dynamics population $\mathcal{G}^{(0)}$ with random symbolic expressions;
2: **for** $i = 1, 2, \cdots, M$ **do**
3:      Compute $d(\mathcal{F}^{(i-1)})$ and $d(\mathcal{G}^{(i-1)})$ using (5);
4:      **if** $d(\mathcal{F}^{(i-1)}) > d(\mathcal{G}^{(i-1)})$ **then**
5:          Calculate the fitness $\mathsf{f}_F$ of each expression $F$ in $\mathcal{F}^{(i-1)}$ using (6);
6:          **if** $\exists F \in \mathcal{F}^{(i-1)}, \mathsf{f}_F \le \epsilon$, **break**;
7:          Select, cross, and mutate the expressions in $\mathcal{F}^{(i-1)}$ to generate the next population $\mathcal{F}^{(i)}$;
8:          $\mathcal{G}^{(i)} = \mathcal{G}^{(i-1)}$;
9:      **else**
10:         Calculate the fitness $\mathsf{f}_G$ of each expression $G$ in $\mathcal{G}^{(i-1)}$ using (7);
11:        **if** $\exists G \in \mathcal{G}^{(i-1)}, \mathsf{f}_G \le \epsilon$, **break**;
12:        Select, cross, and mutate the expressions in $\mathcal{G}^{(i-1)}$ to generate the next population $\mathcal{G}^{(i)}$;
13:        $\mathcal{F}^{(i)} = \mathcal{F}^{(i-1)}$;
14:      **end if**
15: **end for**
16: $F^* = \arg\min_{F \in \mathcal{F}^{(i)}} \sum_{v \in V, t \in T} e(\int_0^t F(\mathbf{x}_v) + \sum_{u \in N_v} G(\mathbf{x}_v, \mathbf{x}_u) dt, f_\theta \left( \mathsf{G}, \mathbf{X}(t_0), t \right)_v)$;
17: $G^* = \arg\min_{G \in \mathcal{G}^{(i)}} \sum_{v \in V, t \in T} e(\int_0^t F(\mathbf{x}_v) + \sum_{u \in N_v} G(\mathbf{x}_v, \mathbf{x}_u) dt, f_\theta \left( \mathsf{G}, \mathbf{X}(t_0), t \right)_v)$;
18: **return** $F^*, G^*$.

---

### 4.3 COMPARISON

We compare SymDL (Cranmer et al., 2020), NASSymDL (Shi et al., 2023), D-CODE (Qian et al., 2022), TP-SINDy (Gao & Yan, 2022) and PI-NDSR in Table 1. These methods cater to different problem settings, utilizing distinct forms of input and output. SymDL and NASSymDL perform general symbolic regression, finding a function $y = f(x)$ from input-output pairs $(x_i, y_i)_{i=1}^N$. D-CODE focuses on *symbolic regression of dynamics*, taking a single trajectory $\{x(t) | t \in \mathcal{T}\}$ to output the governing ODE $\dot{x} = \mathrm{d}x/\mathrm{d}t$. TP-SINDy and PI-NDSR target *symbolic regression of complex network dynamics*, using multiple trajectories $\{\mathbf{X}(t) | t \in \mathcal{T}\}$ to output symbolic network dynamics $F$ and $G$.

Table 1: Comparison with different methods for symbolic regression. (GN: graph network, NAS: neural architecture search, GS: genetic search)

| Category | Design | SymDL | NASSymDL | D-CODE | TP-SINDy | PI-NDSR |
|---|---|---|---|---|---|---|
| Input | | input-output pairs | input-output pairs | single trajectory | multiple trajectories | multiple trajectories |
| Proxy model | Model design | GN w/ inductive bias | NAS | any regressor | – | PIND |
| | Dynamics fitting data | estimated derivatives | estimated derivatives | raw observations | – | raw observations |
| Formula regression | Prior knowledge | elementary operation | elementary operation | elementary operation | function library | elementary operation |
| | Method | GS | GS | GS | sparse regression | coordinated GS |
| | Supervision | internal functions | internal functions | interpolated trajectory | estimated derivatives | network ref & interp. trajectories |
| Output | | input-output mapping | input-output mapping | ODE | Graph ODE | Graph ODE |

We compare these algorithms in two aspects: proxy models and formula regression. 'Proxy models' are trained to fit data and serve as a basis for deriving symbolic expressions. 'Formula regression' directly extract symbolic expressions from raw data or proxy models. For proxy model, SymDL uses graph networks (GN) with inductive bias, and NASSymDL employs neural architecture search (NAS) for skeleton search. D-CODE can incorporate any suitable regressor. PI-NDSR fits multiple trajectories using PIND, a graph neural ODE aligned with network dynamics for better generalization. SymDL and NASSymDL rely on potentially noisy estimated derivatives for dynamics fitting, whereas D-CODE and PI-NDSR train directly on raw observations for improved accuracy. TP-SINDy operates without a proxy model.

In formula regression, methods using genetic search employ elementary operations (e.g., $+, -, \times, \div, \sin, \exp$) to represent formula, offering more flexibility and requiring less prior knowledge than TP-SINDy's linear combination of functions in predefined function library. SymDL and NASSymDL use the internal functions (Cranmer et al., 2020) from the proxy models as supervision to compute fitness in genetic search. D-CODE uses the interpolated trajectories as supervision. TP-SINDy is based on sparse regression and uses estimated derivatives for symbolic regression, which can be noisy and inaccurate over large time intervals. PI-NDSR design a coordinated genetic search, using (4) from proxy model as references and interpolated trajectories as supervision.

## 5 EXPERIMENTS

In this section, we conduct experiments on both synthetic and real-world datasets to evaluate the PI-NDSR. All experiments are implemented with PyTorch (Paszke et al., 2019), PyTorch Geometric (Fey & Lenssen, 2019), and gplearn (Stephens, 2015) in NVIDIA GeForce RTX 4090 GPUs and AMD EPYC 7763 Processors.

### 5.1 EXPERIMENTS ON SYNTHETIC DATASET

**Baseline** We compare our method with baselines SymDL (Cranmer et al., 2020), SINDy (Brunton et al., 2016) and Two-Phase SINDy (TP-SINDy)(Gao & Yan, 2022). Cranmer et al. (2020) cannot be applied directly to symbolic regression of dynamics, we adopt it as a baseline by first using numerical methods (the same methods as TP-SINDy) to estimate derivatives and then applying their method. SINDy (Brunton et al., 2016) is a sparse regression methods to find symbolic dynamics. SINDy first numerically estimates the derivative of each node's activity through the five-point approximation (Sauer, 2011) and then optimizes the coefficients of the linear combination of predefined orthogonal basis functions. TP-SINDy is an improved version of it, which contains more elementary functions and a extra finetuning phase to remove terms with small coefficients. NASSymDL (Shi et al., 2023) is not compared because it cannot find a better network architecture than PIND. We also

do not compare against D-CODE (Qian et al., 2022) because it does not have a natural extension to the dynamics regression of multiple trajectories.

**Dataset**   We investigate the following four network dynamics in experiments, i.e., Susceptible-Infected-Susceptible (SIS) Epidemics Dynamics (Pastor-Satorras et al., 2015), Lotka-Volterra (LV) Population Dynamics (MacArthur, 1970), Wilson-Cowan Neural Firing Dynamic (Lau-

Table 2: Dynamics for generating synthetic dataset.

|     | node dynamics | edge dynamics |
| --- | --- | --- |
| SIS | $-\delta x_i(t)$ | $(1 - x_i(t))x_j(t)$ |
| LV  | $x_i(t)(\alpha - \theta x_i(t))$ | $-x_i(t)x_j(t)$ |
| WC  | $-x_i(t)$ | $(1 + \exp(-\tau(x_j(t) - \mu)))^{-1}$ |
| KUR | $\omega$ | $\sin(x_i(t) - x_j(t))$ |

rence et al., 2019; Wilson & Cowan, 1972) and Kuramoto Oscillators(KUR) model (Kuramoto & Kuramoto, 1984). Their dynamics are shown in Table 2. We conduct experiments on two complex network structures, i.e., Erdős-Rényi (ER) graph (Erdos & Renyi, 1959) and Barabási-Albert (BA) graph (Barabási & Albert, 1999) with 200 nodes.

We randomly initialize the state of all nodes and regularly sample $K$ timestamps $t_0, t_1, \cdots, t_{K-1}$ from the range $[0, T]$ because all other baselines rely on the equal time interval to estimated time derivatives. Then we simulate the whole dynamics to get the node states $[\mathbf{X}(t_0), \mathbf{X}(t_1), ..., \mathbf{X}(t_{K-1})]$. The edge weight $a_{vu}$ is set to binary values, i.e., $a_{vu} = 1$ if there is an edge between node $v$ and node $u$, otherwise $a_{vu} = 0$.

**Evaluation metrics**   The performance is evaluated by two metrics. (a) The recovery probability (**Rec. Prob.**) of formulas with correct skeletons. (See Appendix A for computation details). (b) The mean squared error (**MSE**) between the simulated trajectories using the recovered symbolic expression of dynamics and the ground truth observations. For the fair evaluation, we only compute MSE for the symbolic expressions with correct skeletons. So the MSE can measure how accurate the constants in the formulas are.

**Results**   The comparison results are shown in Table 3. The proposed PI-NDSR generally has a higher recovery probability. For SIS and LV dynamics, TP-SINDy is not stable enough to recover the formula with the correct skeleton. This may result from the instability in the numerical estimation step for derivatives and the failure in narrowing down model space because normalized data may cause the overfitting of candidate functions. For the WC dynamics, the TP-SINDy always fails to regress the correct skeleton, this is because the edge dynamics evolves a parametric function that cannot be represented by a linear combination of predefined functions. For the KUR dynamics, both TP-SINDy and PI-NDSR succeed with recovery probability 1. SINDy does not contain the finetuning phase which TP-SINDy has. As a result, it exhibits a lower recovery probability compared to TP-SINDy. SymDL also relies on the unstable numerical estimation of derivatives, and its symbolic regression process relies solely on the proxy model while PI-NDSR utilizes additional information from the denoised and augmented trajectories. As a result, PI-NDSR achieves the highest recovery probability.

Even in experiments where both methods successfully produce the correct formula skeleton, PI-NDSR consistently achieves better performance. This is because other baselines include a numerical estimation step for derivatives, which can introduce additional errors and result in inaccuracies in the constants of the formula.

## 5.2   EXPERIMENTS ON REAL DATASET

**Dataset**   We demonstrate the effectiveness of PI-NDSR on the real epidemic network. We use the same influenza A (H1N1) spreading dataset as (Gao & Yan, 2022). In this dataset, each node represents a country or region, with the daily counts of newly reported cases serving as the state of these nodes. The edges of the complex network are defined by the global aviation routes, depicting human mobility between regions. Our goal is to uncover the dynamics that govern the spread of the disease. To ensure a fair comparison, we employed the same data preprocessing procedures as (Gao & Yan, 2022), such as constructing the adjacency matrix and cleaning the data.

**Results**   We use PI-NDSR and TP-SINDy to regress the symbolic expression of dynamics for the spreading of influenza A. The symbolic expressions of the spreading of influenza A regressed by

Table 3: Performance comparison on synthetic datasets. MSE values are scaled by $10^{-2}$ and multiply by $10^{-2}$ to obtain the actual values. (TP: TP-SINDy, PI: PI-NDSR, NA: MSE is not applicable because of failure of the correct skeleton recovery.)

| Graphs | Dynamics | Rec. Prob.↑ | | | | MSE↓ ($10^{-2}$) | | | |
|--------|----------|-------|-------|----|----|-------|-------|----|----|
| | | SymDL | SINDy | TP | PI | SymDL | SINDy | TP | PI |
| BA | SIS | 0.35 | 0.11 | 0.15 | 1 | 0.979±0.173 | 0.484±0.056 | 0.434±0.052 | 0.312±0.012 |
| | LV | 0.16 | 0.12 | 0.20 | 1 | 2.075±0.303 | 1.170±0.049 | 0.875±0.057 | 0.136±0.008 |
| | KUR | 0.80 | 0.87 | 1 | 1 | 0.064±0.018 | 0.175±0.016 | 0.040±0.003 | 0.007±0.001 |
| | WC | 0.56 | 0 | 0 | 1 | 0.362±0.057 | NA | NA | 0.092±0.004 |
| ER | SIS | 0.31 | 0.11 | 0.17 | 1 | 1.173±0.095 | 0.468±0.059 | 0.386±0.051 | 0.119±0.025 |
| | LV | 0.15 | 0.09 | 0.19 | 1 | 1.784±0.236 | 0.941±0.041 | 0.763±0.077 | 0.251±0.007 |
| | KUR | 0.87 | 0.78 | 1 | 1 | 0.071±0.018 | 0.087±0.022 | 0.069±0.019 | 0.017±0.001 |
| | WC | 0.40 | 0 | 0 | 1 | 0.266±0.047 | NA | NA | 0.044±0.003 |

PI-NDSR is

$$\dot{\mathbf{x}}_v(t) = a\mathbf{x}_v(t) + \sum_{u \in N_v} \frac{b}{1 + \exp - (m\mathbf{x}_v(t) + c)}\mathbf{x}_u(t), \tag{8}$$

where $a = 0.0740$, $b = 0.0015$, $m = -0.0041$ and $c = 9.9643$. Node dynamics in (8) is a linear function, which aligns with the exponential growth of the epidemic. Edge dynamics in (8) is proportional to the neighboring region's state, which is consistent with the fact that the epidemic spreads increases with the number of infected cases in neighboring regions. The other factor of edge dynamics consists of a composition of a linear transformation followed by a sigmoid activation. This indicates that the infection from neighboring regions may gradually decrease as the number of infected cases in the current region increases. This trend may be caused by the reduction of the willingness of people to travel to epidemic areas or the decrease of basic reproduction number ($R_0$) when the density of infected people is high.

The symbolic expression of the spreading of influenza A regressed by TP-SINDy is

$$\dot{\mathbf{x}}_v(t) = a'\mathbf{x}_v(t) + \sum_{u \in N_v} \frac{b'}{1 + \exp - (\mathbf{x}_v(t) - \mathbf{x}_u(t))}, \tag{9}$$

where $a' = 0.074$ and $b' = 7.130$. (9) fails to capture the spreading pattern of the epidemic. When there are no infected cases in the complex network, the edge dynamics of (9) will still result in a non-zero growth rate, which is not reasonable from a physical perspective. In the same scenario, (8) yields zero growth rate, which is consistent with the fact that the epidemic will not spread when there are no infected cases.

We compare the trajectories of symbolic expressions in (8) and (9) with the real infected cases. Fig. 2 visualizes the simulation results of inferred dynamics in two regions, i.e., "Finland" and "Saint Pierre and Miquelon". The trajectories of the infected cases in the two regions inferred by PI-NDSR are consistent with the ground truth, while the trajectories inferred by TP-SINDy deviate from the ground truth.

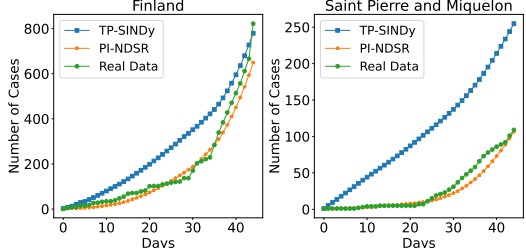

We quantitatively evaluate the errors of regressed dynamics. As the scale of infected cases varies across different regions, we normalize the

Figure 2: Visualizing the predicted number of newly reported cases in two regions using symbolic expressions from TP-SINDy and PI-NDSR.

infected cases to the range of $[0, 1]$ by the maximal value of each region. The mean square error (MSE) of TP-SINDy is $0.9028$, while the MSE of PI-NDSR is $0.8261$. The results show that PI-NDSR fits the neural dynamics better.

## 5.3 ROBUSTNESS ON PI-NDSR AND TP-SINDY

We adopt the evaluation settings in (Qian et al., 2022) to test the robustness of our method. We evaluate the performances when observations are noisy or the time interval is large. We select KUR dynamics to evaluate the robustness of PI-NDSR and TP-SINDy.

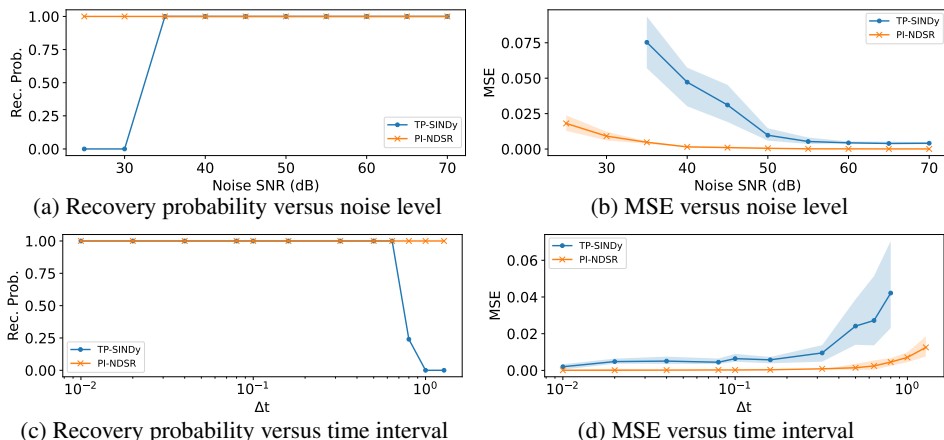

Figure 3: Evaluation of robustness. The shaded areas correspond to 95% confidence interval. (a) and (b) show the recovery probability and MSE when adding noise to the observations. (c) and (d) show the recovery probability and MSE when increasing the time interval between observations.

Firstly, we add Gaussian noise to all node states to evaluate its performance under noise. The magnitude of noise is measured by the signal-to-noise ratio (SNR). The results of recovery probability are presented in Fig. 3(a). In the experiments, as the signal-to-noise ratio (SNR) decreases from 70 dB to 25 dB, our method consistently maintains a 100% recovery rate. In contrast, TP-SINDy's recovery rate drops to 0% when the SNR reaches 30 dB. In Fig. 3(b), the proposed method consistently produces more accurate symbolic expressions that have lower MSE. The reason is that TP-SINDy relies on numerically estimating time derivatives that are noisy and inaccurate. On the other hand, our method uses neural dynamics to denoise and interpolate observations directly. Deep neural networks can handle noisy observations effectively because they have the ability to learn and extract meaningful patterns from large amounts of data, even when the data contains significant noise. Using the accurately denoised observations, PI-NDSR can predict constants in the formula better and produce a more accurate trajectory when noise exists.

TP-SINDy relies on the equal time interval to estimate time derivatives. So we increase the size of time intervals to evaluate the performances of both TP-SINDy and PI-NDSR. We sample the timestamps regularly from $[0, 100]$ with different intervals $\Delta t$. As shown in Fig. 3(c), our model achieved a 100% recovery probability across all values of $\Delta t$, whereas TP-SINDy consistently failed to recover the correct formula skeleton when the time interval is large. Fig. 3(d) shows that PI-NDSR always produces more accurate results when both methods produce the correct skeleton of dynamics. This advantage arises because the interpolated observations in PI-NDSR are better suited when the time interval is large compared to the estimated time derivatives used by TP-SINDy. Visualization of interpolated trajectories and estimated time derivatives are shown in Fig. 5 of the appendix.

## 6 CONCLUSION

In this paper, we propose physically inspired symbolic regression to learn symbolic expressions of complex network dynamics from data. Our approach aims to avoid overfitted symbolic formulas by incorporating the supervision of interpolated and denoised trajectories, as well as referencing neural dynamics. Our method is based on neural dynamics to augment and denoise the trajectory of complex network, and then apply the coordinated genetic search to infer the symbolic expressions based on the dynamics reference from neural dynamics. Compare with existing methods, our method requires less prior knowledge on complex networks, can handle irregularly sampled data, and effectively search the symbolic space and avoid overfitting. See impact statement, future works and limitations in Appendix C.

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

# A DETAILS ON EXPERIMENTS

## A.1 DATASET STATISTICS

The BA graph is generated with 200 nodes and the initial degree of each node is set to 3. The ER graph is generated with 200 nodes and the probability for edge creation is set to 0.02. The initial states of SIS, LV, and WC dynamics are generated by randomly sampling from $[0, 1]$. For KUR dynamics, the initial states are generated by randomly sampling from $[0, 2\pi]$. For SIS dynamics, we set $\delta = 0.5$. For LV dynamics, we set $\alpha = 0.75$, $\theta = 0.5$,. For KUR dynamics, we set $\omega = 0.75$. For WC dynamics, we set $\tau = 0.75$, $\mu = 0.5$. We regularly sample 100 timestamps from $[0, 1]$ and simulate the dynamics to generate the observation data.

## A.2 DETAILS FOR NETWORK TRAINING

We split the timestamps randomly into training, validation, and testing sets with a ratio of 0.8, 0.2, 0.1 to train the NeuralODE. We train the neural dynamics for 1000 epochs using optimizer AdamW. The learning rate is searched in the range of $[1e - 3, 1e - 2]$, the weight decay is set to $0.001$. We use MLPs as the encoder and decoder of neural dynamics. The hidden dimension of the neural dynamics is set to 10. The details of the network structure are shown in Table 4.

Table 4: Details of network structure for different dynamics.

|  | SIS | LV | KUR | WC | real dataset |
|---|---|---|---|---|---|
| Hidden dimension | 10 | 10 | 10 | 10 | 10 |
| Activation of $\phi^n$ | ReLU | ReLU | ReLU | ReLU | Sigmoid |
| Activation of $\phi^e$ | ReLU | Tanh | Tanh | Sigmoid | Sigmoid |
| Activation of Encoder | ReLU | Tanh | Tanh | ReLU | Tanh |
| Activation of Decoder | ReLU | Tanh | Tanh | ReLU | Tanh |
| Layer of $\phi^n$ | 2 | 2 | 1 | 1 | 2 |
| Layer of $\phi^e$ | 2 | 2 | 3 | 2 | 3 |

## A.3 DETAILS OF GENETIC SEARCH

We implement the coordinated genetic search based on gplearn (Stephens, 2015). gplearn (Stephens, 2015) represent the symbolic expressions as a syntax tree, where the functions are interior nodes, and the variables and constants make up the leaves. Evolution such as crossover, mutation, and reproduction are performed on the syntax tree. The population size of $\mathcal{F}$ and $\mathcal{G}$ are set to 200. The maximum generation of the genetic search $M$ is set to 50 and the stopping threshold $\epsilon = 10^{-5}$. The $K$ in Algorithm 1 equals to 20. The function set includes addition, subtraction, multiplication, division, sine, cosine, and exponential. The constants are constrained in the range $[-1, 1]$. Other hyperparameters of gplearn are set as: p_crossover=0.6, p_subtree_mutation=0.1, p_hoist_mutation=0.05, p_point_mutation=0.1, parsimony_coefficient=0.01. We conduct the genetic search in 256 parallel threads to speed up the search process. Our CPUs are two AMD EPYC 7763 Processors.

## A.4 COMPUTATIONAL DETAILS OF REC. PROB.

The recovery probability is calculated as the ratio of the number of successful recovery of formula skeletons to the total number of experiments. We automatically check the correctness of the recovered formula skeletons using the method for verifying skeletons provided in Qian et al. (2022). Basically, we replace the constants in the formulas with placeholders and use the $\mathsf{simplify}(f' - f) == 0$ criterion from the Sympy package to determine if the skeleton is correct.

## A.5 DISCUSSIONS ON THE CHOICE OF METRICS

**Compute MSE between trajectories instead of the constants.** We do not directly compute the MSE between predicted and true constants. This is because our goal is to evaluate how well the obtained symbolic expressions predict trajectories, which is crucial for real-world scenarios like

epidemic forecasting. Directly computing constant errors is insufficient, as different constants impact the trajectory differently. Some constants require high precision, with small deviations causing significant errors, while others are less critical and can tolerate some errors.

**Compute MSE for formulas with correct skeletons.** For simulated datasets, we choose MSE restricted to correctly recovered skeletons because the baseline methods often exhibit large MSE when recovering incorrect skeletons. Filtering out these formulas allows the baselines to achieve comparable performance. For real datasets, since the true dynamics skeleton is unknown, we directly compare the MSE of the trajectories without filtering by the skeletons.

## B ADDITIONAL RESULTS

### B.1 ABLATION STUDY

We conduct ablation studies to demonstrate the necessity of the components in PI-NDSR. We test two variants of PI-NDSR: (a) Using the original observations instead of interpolated and denoised trajectory when doing the coordinated genetic search. (without Interp.) (b) Using the interpolated trajectory but evolving both populations $\mathcal{F}$ and $\mathcal{G}$ every time. (without Coord.)

Table 5 shows the results of SIS and LV dynamics in the BA graph. Without the interpolated and denoised observations, both the recovery probability and the accuracy of PI-NDSR drop. This indicates that the interpolated and denoised trajectories can provide high-quality supervision data for symbolic regression. Without the coordinated genetic search, the recovery probability of PI-NDSR drops significantly, and when the skeleton can be successfully recovered the MSE increases slightly, indicating that the search strategy can avoid overfitted expressions in symbolic space and help find correct dynamics expressions.

Table 5: Ablation study with experiment results on SIS and LV dynamics in BA graph.

| Model | SIS | | LV | |
|---|---|---|---|---|
| | Rec. Prob.↑ | MSE↓ ($10^{-2}$) | Rec. Prob.↑ | MSE↓ ($10^{-2}$) |
| PI-NDSR | 1 | 0.312±0.012 | 1 | 0.136±0.008 |
| PI-NDSR(w/o Interp.) | 0.81 | 0.408±0.027 | 0.86 | 0.588±0.028 |
| PI-NDSR(w/o Coord.) | 0.31 | 0.326±0.015 | 0.47 | 0.142±0.014 |

We also experiment on the robustness of the ablation variants. Table 6 shows the results of KUR dynamics in the BA graph when the observations are noisy or the time interval is large. Different from the results in Table 5, the success Prob. significantly drop when removing the interpolation part. This proves the effectiveness of neural dynamics in denoising and augmenting trajectories.

Table 6: The robustness of two variants compared with full method on KUR dynamics in BA graph.

| Models | Noise (SNR=35dB) | | Time interval ($\Delta t = 1.28$) | |
|---|---|---|---|---|
| | Rec. Pro.↑ | MSE↓ ($10^{-2}$) | Rec. Pro.↑ | MSE↓ ($10^{-2}$) |
| PI-NDSR | 1 | 0.478±0.103 | 1 | 0.454±0.213 |
| PI-NDSR(w/o Interp) | 0.84 | 6.970±1.870 | 0.78 | 2.645±1.138 |
| PI-NDSR(w/o Alter) | 0.76 | 0.504±0.094 | 0.66 | 0.570±0.241 |

### B.2 RUNTIME

In Table 7, PI-NDSR saves 30.1% running time on SIS dynamics and 39.0% running time on LV dynamics compared with PI-NDSR(w/o Alter). The results show that the coordinated genetic search can significantly reduce the search space and improve the efficiency of the search process.

Table 7: The runtime (minutes) of PI-NDSR and PI-NDSR(w/o Coord.).

| Model | SIS | LV |
|---|---|---|
| PI-NDSR | 61.5 | 50.9 |
| PI-NDSR(w/o Coord.) | 88.0 | 83.4 |

## B.3 EXAMPLES OF EXPRESSIONS FROM SYMBOLIC EXPRESSION

In this section, we provide examples of symbolic expressions of PI-NDSR, TP-SINDy (Rec.), and TP-SINDy (Fail) on SIS, LV, KUR, and WC dynamics in the BA graph. TP-SINDy (Rec.) represents the symbolic expressions of TP-SINDy when the skeleton of the dynamics is successfully recovered, while TP-SINDy (Fail) represents the symbolic expressions of TP-SINDy when the skeleton of the dynamics is not successfully recovered. The expressions are shown in Table 8.

Table 8: Symbolic regressions of PI-NDSR, TP-SINDy (Rec.), and TP-SINDy (Fail) on SIS, LV, KUR, and WC dynamics in the BA graph.

| Dynamics | Models | Node dynamics | Edge dynamics |
|---|---|---|---|
| SIS | GT | $-0.5x_i(t)$ | $(1 - x_i(t))x_j(t)$ |
| | PI-NDSR | $-0.48540x_i(t)$ | $(1 - x_i(t))x_j(t)$ |
| | TP-SINDy (Rec.) | $-0.46640x_i(t)$ | $(0.99119 - 1.09637x_i(t))x_j(t)$ |
| | TP-SINDy (Fail) | $-0.47256x_i^2(t) - 0.12596$ | $0.10860\mathrm{sigmoid}(x_j(t) - x_i(t)) +$ $0.18835(x_j(t) - x_i^2(t))$ $0.19917(x_j(t) - x_i(t)) + 0.35416\sin(x_j(t))$ |
| LV | GT | $x_i(t)(0.75 - 0.5x_i(t))$ | $-x_i(t)x_j(t)$ |
| | PI-NDSR | $x_i(t)(0.75034 - 0.48812x_i(t))$ | $-0.99428x_i(t)x_j(t)$ |
| | TP-SINDy (Rec.) | $x_i(t)(0.69882 - 0.41853x_i(t))$ | $-0.91701x_i(t)x_j(t)$ |
| | TP-SINDy (Fail) | $0.03984 + 0.36330 * \sin(x_i(t))$ | $-0.945810x_i(t)x_j(t) - 0.11895x_i(t)x_j^2(t)$ |
| KUR | GT | $0.75$ | $\sin(x_i(t) - x_j(t))$ |
| | PI-NDSR | $0.75002$ | $\sin(1.0001x_i(t) - x_j(t))$ |
| | TP-SINDy (Rec.) | $0.75014$ | $0.99899\sin(x_i(t) - x_j(t))$ |
| | TP-SINDy (Fail) | NA | NA |
| WC | GT | $-x_i(t)$ | $\mathrm{sigmoid}(-0.75(x_j(t) - 0.5))$ |
| | PI-NDSR | $-x_i(t)$ | $\mathrm{sigmoid}(-0.74503(x_j(t) - 0.49128))$ |
| | TP-SINDy (Rec.) | NA | NA |
| | TP-SINDy (Fail) | $-0.82267x_i(t)$ | $0.08513\mathrm{sigmoid}(x_j(t) - x_i(t))$ $+0.68484\mathrm{sigmoid}(x_j(t))$ |

## B.4 EXAMPLE OF OVERFITTING

Take the SIS dynamics in the BA graph as an example. The symbolic expressions of PI-NDSR, TP-SINDy (Rec.), and TP-SINDy (Fail) are shown in Table 8. We compute the MSE of the predicted trajectories under interpolated and extrapolated settings. The results are shown in Table 9. Although the symbolic expressions of TP-SINDy (Rec.) and TP-SINDy (Fail) have relatively low MSE values under the interpolated setting, their MSE values are much higher under the extrapolated setting. This indicates that the symbolic expressions of TP-SINDy are overfitted and cannot generalize well to extrapolated setting. We double the time range from $[0, 1]$ to $[0, 2]$ to evaluate the extrapolation performance.

Note that all failure cases in Table 8 can also be viewed as examples of overfitting. The symbolic expressions of PI-NDSR are more interpretable and simpler while the overfitted symbolic expressions of TP-SINDy are more complex and contain more terms.

Table 9: The MSE of PI-NDSR, TP-SINDy (Rec.), and TP-SINDy (Fail.) under interpolated and extrapolated settings on SIS dynamics in the BA graph.

|  | PI-NDSR | TP-SINDy (Rec.) | TP-SINDy (Fail.) |
|---|---|---|---|
| Interpolation | $3.2 \times 10^{-3}$ | $5.2 \times 10^{-3}$ | $147.2 \times 10^{-3}$ |
| Extrapolation | $3.9 \times 10^{-3}$ | $46.9 \times 10^{-3}$ | $697.0 \times 10^{-3}$ |

### B.5 VISUALIZATION OF NEURAL DYNAMICS

**Node/edge dynamics estimation** With the inductive bias, the neural node dynamics and edge dynamics computed with (4) are accurate enough to be used as the neural references. We visualize the result for LV dynamics in the BA graph in Fig. 4. It can be seen that the neural estimations of dynamics are very close to the ground truth.

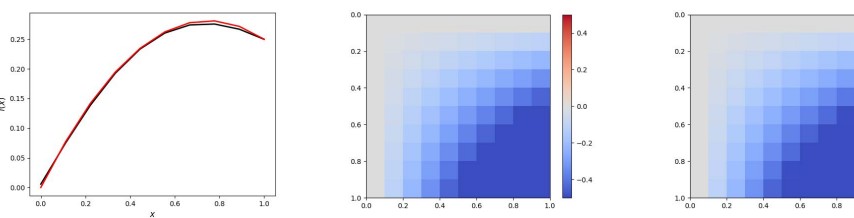

(a) Node dynamics. $\hat{F}$ is in red, ground truth is in black. (b) Ground truth of edge dynamics. (c) Neural estimation of edge dynamics.

Figure 4: Visualization of neural dynamics estimation for LV dynamics in the BA graph. For node dynamics, we show the values at intervals of 0.1 between 0 and 1. For edge dynamics, we use a heatmap to show the values in (b) and (c).

**Observation denoising and interpolating** When observations are noisy or time interval is large, the neural dynamics can denoise and interpolate the observations to provide high-quality supervision data for symbolic regression. On the other hand, the numerical estimation is sensitive to noise and needs the sample interval to be small enough. We visualize the interpolate trajectories and the estimated time derivatives in Fig. 5, which is consistent with our contributions.

### B.6 RESULTS FOR MULTI-DIMENSIONAL DYNAMICS

The propose method can be applied to multi-dimensional dynamics. We test the performance of PI-NDSR on the FitzHugh–Nagumo (FHN) dynamics which are proposed to model the activity of neural systems (Rabinovich et al., 2006). The formula is shown in Table 10. The dimension 1 represents the membrane voltage and dimension 2 represents the recovery variable.

Table 10: Dynamics for FitzHugh-Nagumo dynamics.

|  | node dynamics | edge dynamics |
|---|---|---|
| dimension 1 | $x_{i,1}(t) - x_{i,2}(t) - \frac{1}{3}x_{i,1}(t)^3$ | $x_{j,1}(t) - x_{i,1}(t)$ |
| dimension 2 | $ax_{i,1}(t) + bx_{i,2}(t) + c$ | $0$ |

The neural network can be directly applied to multi-dimensional dynamics by extending its input dimensions. For the genetic search component, we adapt the existing package to support vector-valued functions. Gplearn (Stephens, 2015) represents scalar-valued functions using a syntax tree. In our approach, vector-valued functions are represented as a "syntax forest," which is a collection of syntax trees. Mutation and crossover operations are conducted independently for each dimension. This coordinated genetic search framework seamlessly extends to handle multi-dimensional dynamics.

We evaluate the performance of PI-NDSR on the FHN dynamics within a BA graph. PI-NDSR successfully reconstructs the dynamics' skeleton with a success probability of 1. Furthermore, it

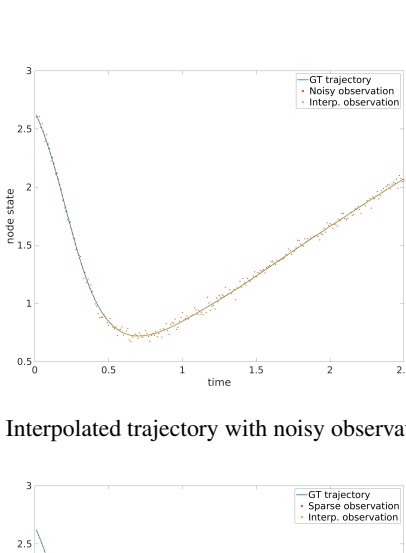

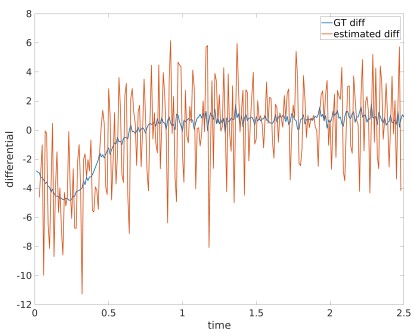

(a) Interpolated trajectory with noisy observation.

(b) Estimated time derivative with noisy observation.

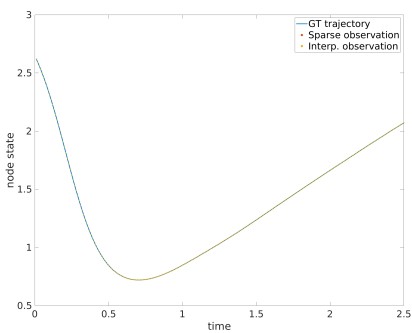

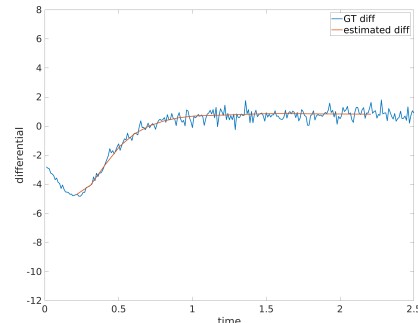

(c) Interpolated observations with large sample time interval.

(d) Estimated time derivative with large sample time interval.

Figure 5: Visualization of interpolated and denoised observations and the estimated time derivative. (a) The interpolated observations are very close to the ground truth when noise exists. (b) The estimated time derivative is inaccurate with noisy observation. (c) The interpolated observations are close to the ground truth with a large time interval (0.1). (d) The estimated time derivative is inaccurate when the sample time interval is large.

achieves a mean squared error (MSE) of $0.182 \times 10^{-2}$, outperforming TP-SINDy, which yields a higher MSE of $0.454 \times 10^{-2}$.

## C   DISCUSSIONS

**Motivation of neural dynamics design**   The rationale for decoupling node and edge dynamics is motivated by Xu et al. (2019) which study the reasoning ability of DNN. This paper provides a mathematical framework explaining why certain architectures may be more suitable for particular reasoning tasks. The primary theorem (Theorem 3.6 of Xu et al. (2019)) asserts that improved algorithmic alignment (Definition 3.3 of Xu et al. (2019), structural correspondence between functions of a reasoning task and different modules in DNN) leads to enhanced generalization.

PI-NDSR learns neural dynamics with strong generalization ability. Based on algorithmic alignment, (1) provides a strong inductive bias for the neural dynamics design. The designed model (2) is algorithmically aligned with (1), allowing our model to demonstrate strong generalization. With the strong generalization, the decoupling of node and edge dynamics can provide a better reference for coordinated genetic search algorithm to recover symbolic expressions of complex network dynamics. For example, the node dynamics can be seen as the dynamics on a complex network with a single node.

**Impact statement**   The proposed method can be applied to various real-world scenarios, such as epidemic forecasting, brain dynamics, and single-cell RNA forecasting, helping understand the underlying mechanisms of various complex systems.

**Limitations**   Our method has several limitations:

- PI-NDSR cannot successfully recover symbolic expressions when the formulations are highly complex or the dimension of node states is high. The highly complex formulations indicate a large search space for the genetic algorithm. Therefore, there should be a large population size and a large number of generations for the genetic algorithm to find the symbolic expressions. Since the fitness of our method is calculated based on the pairwise combination of node and edge dynamics, the fitness evaluation is computationally expensive and memory-consuming.
- PI-NDSR cannot deal with the complex network dynamics when some variables are missing in the observations. In some complex systems, it is difficult to observe all variables at the same time. In this case, the prediction accuracy of neural dynamics (3) may not be high enough to provide high-quality supervision data for symbolic regression.

**Future works**   In the future, we plan to extend our method to handle more complicated network dynamics with impressive formula manipulation capability of LLM, which includes dynamics involving time delays, multiple variables, and missing variables. Additionally, we are interested in exploring the application of our method in various real-world scenarios like brain dynamics and single-cell RNA forecasting.

