# OpenReview forum: "Neural Symbolic Regression of Complex Network Dynamics"
_ICLR.cc/2025/Conference — Submitted to ICLR 2025_

### Official Review · Reviewer_KBLS · 2024-11-01

**Soundness:** 2
**Presentation:** 3
**Contribution:** 2
**Rating:** 5
**Confidence:** 4

**Summary:**

This paper introduces Physically Inspired Neural Dynamics Symbolic Regression (PI-NDSR), a technique that combines neural networks with genetic programming to autonomously discover symbolic representations of dynamics. PI-NDSR includes two main components: a Physically Inspired Neural Dynamics (PIND) module, which enhances and denoises trajectories
through observed trajectory interpolation; and a coordinated genetic search algorithm that extracts symbolic expressions. This algorithm uses references from both node and edge dynamics provided by neural dynamics to prevent overfitting in symbolic space.

**Strengths:**

1. The author proposes a network dynamics symbolic regression method that uses neural networks to complete data, which can help mitigate the problem of noisy data trajectories.
2. By applying genetic programming for symbolic regression, this method avoids the need for a pre-designed knowledge base.

**Weaknesses:**

1. The innovation of the method in the article is limited. First, the use of neural networks to fit data in order to enhance symbolic regression performance has been widely adopted in many works. For instance, in AI-Feynman[1], a neural network is first used to fit the data, and then the fitted data is analyzed to infer the nature of the equation, which facilitates subsequent symbolic regression. Similarly, this paper also uses a neural network to fit data to avoid inaccuracies in derivative estimation caused by sparse or noisy observational data. Essentially, both approaches leverage the generalization capability of neural networks to mitigate the negative impact of sparse data points. Second, using surrogate models for symbolic regression is also a common approach. For example, SymDL [2], like this paper, uses an MLP locally as a surrogate model for node dynamics G and edge dynamics F.
2. The experiments in the article are not sufficiently comprehensive; they are limited to relatively simple dynamical scenarios and have not been applied to more complex scenarios such as neuronal, genetic, social, and coupled oscillator dynamics. Moreover, the experiments are restricted to one-dimensional data, while many other works, like TP-SINDy, can handle higher-dimensional data. For example, the FHN dynamics is a higher-order network equation, and this paper does not include experimental discussion on this equation.


References:
[1] Silviu-Marian Udrescu, Max Tegmark ,AI Feynman: A physics-inspired method for symbolic regression.Sci. Adv.6,eaay2631(2020).DOI:10.1126/sciadv.aay2631.
[2] Cranmer M, Sanchez Gonzalez A, Battaglia P, et al. Discovering symbolic models from deep learning with inductive biases[J]. Advances in neural information processing systems, 2020, 33: 17429-17442.
[3] Gao, TT., Yan, G. Autonomous inference of complex network dynamics from incomplete and noisy data. Nat Comput Sci 2, 160–168 (2022). https://doi.org/10.1038/s43588-022-00217-0.

**Questions:**

1. How does the computational efficiency of GA compare to TP-SINDy in the symbolic regression component? Please include an appropriate discussion on this part. If the method in this paper takes too long, using a predefined library, as in TP-SINDY, remains a good option. Determining which approach is better requires evaluating the time consumption of each method.

---

> ### Author Response · Authors · 2024-11-25
> **Rebuttal by Authors**
>
> We thank the reviewer for valuable comments and suggestions. We respond to the concerns raised by the reviewer as follows:
>
> ### Weaknesses
>
> **W1:** limited innovation because of similarity to AI-Feynman and SymDL
> > **R1:** Our paper introduces a novel coordinated genetic search algorithm that leverages the strengths of both neural network interpolation and surrogate model referencing to uncover symbolic expressions representing complex network dynamics. While neural network interpolation and surrogate model referencing have been utilized separately in methods such as AI-Feynman and SymDL, our approach uniquely integrates these components in a synergistic manner. This integration, achieved through our coordinated genetic search algorithm, is the primary innovation of our work.
>
>
> **W2:** more complex scenarios such as neuronal, genetic, social, and coupled oscillator dynamics & multi-dimensional dynamics
> > **more complex scenarios:** The dynamics in our current experiments already encompass the scientific domains you mentioned. For example, the Wilson-Cowan Neural Firing Dynamics and the FitzHugh-Nagumo Dynamics (newly added in Appendix B.6 of the revision) represent complex dynamics in neuroscience. The Lotka-Volterra Population Dynamics are applied in modeling genetic evolution, while the Susceptible-Infected-Susceptible Epidemics Dynamics are used by social scientists to model the spread of rumors. Kuramoto Oscillators are a type of coupled oscillator dynamics.
> >
> > **Multi-dimensional network dynamics:** We add a two-dimensional network dynamics experiment of the FHN model in Appendix B.6 of the revision. The proposed method can handle the multi-dimensional network dynamics after extending the existing genetic search library, gplearn, to support vector-valued functions.
> ### Questions
>
> **Q1:** computational efficiency of GA compared to TP-SINDy in the symbolic regression component
> > **A1:** We have reported the running time of our method in Appendix B.2. While TP-SINDy completes in approximately 4 minutes, PI-NDSR requires 50 to 60 minutes. Although running time is not the primary strength of our method, this should not be considered a drawback. The main objective of symbolic regression is to uncover and better understand the underlying dynamics of a system, where accuracy is of utmost importance. Sacrificing some computational efficiency for more accurate symbolic expressions is a worthwhile trade-off.

---

> > ### Comment · Reviewer_KBLS · 2024-11-27
> >
> > Thanks for the authors' response. Although the authors have addressed my concerns by adding experiments and discussions, the proposed method is essentially a combination of neural network interpolation and surrogate modelling, which makes the novelty of the paper limit. Therefore, I maintain my original score.

---

### Official Review · Reviewer_FzQZ · 2024-11-02

**Soundness:** 2
**Presentation:** 2
**Contribution:** 2
**Rating:** 5
**Confidence:** 4

**Summary:**

This paper proposes a physically-inspired neural dynamics symbolic regression (PI-NDSR) to attempt to address the limitations of sparse regression methods for nonlinear network dynamics that require finite differences and expert design function libraries. It comprises two principal components: a physically-inspired neural dynamic model that enhances trajectories and reduces noise by observing trajectory interpolation; and a coordinated genetic search algorithm to derive symbolic expressions.

**Strengths:**

(1)The proposed method can handle observation data with noise for inferring interpretable network dynamics.
(2)The proposed method circumvents the need for expert knowledge elaboration, necessitating only the basic operator regression symbolic model.

**Weaknesses:**

(1)The rationality of the method design needs to be further clarified. The design and assumptions of the encoder and decoder especially need to be explained. When training the neural network, the input of the Dec in Eq. 2 is \phi^n + \phi^e, but when inferring equations, the input of the Dec becomes \phi^n or \phi^e. This means that the Dec needs to satisfy Dec(\phi^n + \phi^e)=Dec(\phi^n)+Dec(\phi^e), which is a strong assumption for the design of the decoder, thus affecting the corresponding inverse operation, i.e. the encoder, which also satisfies the corresponding assumption. There is reason to doubt the effectiveness and rationality of the encoder and decoder.
(2)The contribution of the proposed method lacks clear clarification.
(3)The experimental analysis is not comprehensive. For example, there is a lack of time cost analysis while time consumption is an important metric for algorithm performance evaluation and the states of all network dynamics evaluated in the paper are one-dimensional.

**Questions:**

-Please clarify the design rationale for the encoder and decoder, and explain how to ensure the decoder maintains consistency between training and inference. I suggest that the authors provide empirical evidence or theoretical justification for the effectiveness of the encoder-decoder design.

-Can the method proposed in the paper recover the symbolic expressions of network dynamics when the state of nodes has multiple features? Please either provide results for multi-dimensional node states, or discuss the potential challenges and necessary modifications for extending the method to such cases.

-As described in line 749 of the appendix, Rec.Prob is computed by “simplify,” which suggests that Rec.Prob takes the value 0 or 1. Why is there a float number in Table 3? Please clarify how to calculate the Rec.Prob values in Table 3, and explain why some values are not binary if the calculation method described in the appendix is correct.

-What is the specific meaning of the "error" in line 215 and how is it calculated?

-Lack of discussion on highly relevant work. DNND [1] employs the GNN proxy model to decompose the dynamics and then uses the SINDy to obtain the symbolic expression. Please explain the difference between the method in the paper and DNND and suggest adding comparisons. Please include a comparison with DNND in experiments, or at minimum, to provide a detailed discussion of how the proposed method differs from and potentially improves upon DNND.

-Due to the method's design in this paper being similar to the graph ODE models, the proposed method appears to be able to handle irregularly-sampled data, which could have been considered an advantage but was not mentioned and validated in this paper. I suggest that the authors conduct additional experiments with irregularly-sampled data to validate this capability, or discuss why this potential advantage was not explored in the current work.

-There are some typos and unclear points as follows:
    The inputs in Eq. (2) and Eq. (4) are inconsistent.
    In Eq. (6) and (7), too many G's appear, and these G's seem to have different meanings.

[1] Liu, B., Luo, W., Li, G., Huang, J., Yang, B.: Do we need an encoder-decoder to model dynamical systems on networks? In: Proceedings of the Thirty-Second International Joint Conference on Artificial Intelligence, pp. 2178–2186 (2023)

---

> ### Author Response · Authors · 2024-11-25
> **Rebuttal by Authors (Part 1)**
>
> We thank the reviewer for valuable comments and suggestions. We respond to the concerns raised by the reviewer as follows:
>
> ### Weaknesses
>
> **W1:** rationality of the method design & strong assumption of decoder
> > **R1:**
> >
> > **Rationality of the method design:** The design of the encoder and decoder is motivated by neural algorithmic reasoning [1]. In this paper, we regard the dynamics of the network as an algorithmic reasoning task, i.e., an algorithm with an ODE. So we incorporate various techniques, i.e., algorithmic alignment [1] and encoder-decoder paradigm, into the neural network design.
> >
> > **Strong assumption of decoder:**
> > - **MLP is expressive enough to satisfy the assumption:** The decoder's additivity assumption can be met by a general MLP since the input and output of the decoder are confined to some specific subspace. Within this subspace, the additivity assumption is relatively weak, allowing the decoder to be trained accordingly.
> > - **MLP satisfies the assumption empirically:** We conduct experiments to show that the decoder in our model can satisfy the assumption empirically with SIS dynamics. Since the states of SIS dynamics are bounded in range $[0,1]$, we randomly sample two values $x_1$ and $x_2$ from $[0,1]$ and map them to the hidden representation $\mathbf{h}_1$ and $\mathbf{h}_2$ with encoder. Then we calculate the MSE between $\mathsf{Dec}(\mathbf{h}_1 + \mathbf{h}_2)$ and $\mathsf{Dec}(\mathbf{h}_1) + \mathsf{Dec}(\mathbf{h}_2)$. The result is $0.0021$, which is very small and means that the decoder can satisfy the assumption.
> >
> > [1] https://thegradient.pub/neural-algorithmic-reasoning
> >
> > [2] What Can Neural Networks Reason About?
>
> **W2:** The contribution lacks clear clarification.
> > **R2:** Our contributions are summarized in the following two points:
> > - Using both the interpolated denoised trajectories and references of node/edge dynamics from physically inspired neural dynamics for supervision, PI-NDSR proposes a coordinated genetic search algorithm to avoid overfitted expressions and improve the quality of symbolic expressions.
> > - Compared to previous methods, PI-NDSR has better recovery probability and smaller errors in the recovered symbolic dynamics.
> >
> > To clarify the contribution, we conclude the contribution of our work at the end of the introduction of the revision.
>
> **W3:** time cost analysis & one-dimensional dynamics.
> > **R3:**
> > **Time cost:** We have reported the running time of our method in Appendix B.2. While TP-SINDy completes in approximately 4 minutes, PI-NDSR requires 50 to 60 minutes. Although running time is not the primary strength of our method, this should not be considered a drawback. The main objective of symbolic regression is to uncover and better understand the underlying dynamics of a system, where accuracy is of utmost importance. Sacrificing some computational efficiency for more accurate symbolic expressions is a worthwhile trade-off.
> >
> > **Multi-dimensional network dynamics:** We add a two-dimensional network dynamics experiment of the FHN model in Appendix B.6 of the revision. The proposed method can handle the multi-dimensional network dynamics after extending the existing genetic search library, gplearn, to support vector-valued functions.

---

> > ### Author Response · Authors · 2024-11-25
> > **Rebuttal by Authors (Part 2)**
> >
> > ### Questions
> >
> > **Q1:** Please clarify the design rationale for the encoder and decoder, and explain how to ensure the decoder maintains consistency between training and inference.
> > > **A1:** We explain the design rationale for the encoder and decoder in **R1**. Empirically, the consistency between training and inference can be maintained with end-to-end training according to the experimental results. Please note that the network design part is not the main contribution of our work, our network design of PI-NDSR is good enough for performing symbolic regression.
> >
> > **Q2:** multi-dimensional network dynamics
> > > **A2:** See **R3**.
> >
> > **Q3:** As described in line 749 of the appendix, Rec. Prob is computed by “simplify,” which suggests that Rec. Prob takes the value 0 or 1. Why is there a float number in Table 3? Please clarify how to calculate the Rec. Prob values in Table 3, and explain why some values are not binary if the calculation method described in the appendix is correct.
> > > **A3:** Every single experiment will give a boolean result, i.e., whether the skeleton of the symbolic expression is correctly recovered or not. The Rec. Prob is the ratio of the number of correctly recovered symbolic expressions to the total number of experiments, which is a float number.
> >
> > **Q4:** What is the specific meaning of the "error" in line 215 and how is it calculated?
> > > **A4:** We compute the l1 distance between the predicted trajectory and the ground truth trajectory. Thank you for pointing this out and we have updated our manuscript to clarify this.
> >
> > **Q5:** lack of discussion on highly relevant work DNND[3], which employs the GNN proxy model to decompose the dynamics and then uses the SINDy to obtain the symbolic expression. Please include a comparison with DNND in experiments, or explain the difference.
> >
> > [3] Do we need an encoder-decoder to model dynamical systems on networks?
> > > **A5:**
> > >
> > > **Problem setup of DNND:**
> > > DNND is relevant to our method. We have discussed it in the related work of revision. However, DNND is different from PI-NDSR in the following aspects:
> > > - DNND is proposed to improve the neural dynamics for long-term prediction. PI-NDSR focuses on the symbolic regression of network dynamics. The problem setup of these two papers is different.
> > > - DNND is a neural network designed for improving long-term prediction performance. PI-NDSR is a method that combines neural dynamics and coordinated genetic search, specifically designed for symbolic regression.
> > >
> > > **Difficulty in Quantitative Comparison with DNND:** We have reviewed DNND's paper and discovered that it does not provide a thorough explanation of symbolic regression. It merely mentions it as a potential future endeavor in the conclusion, without any technical implementation. Furthermore, DNND does not offer open-source code. As a result of these missing implementation details and the absence of open-source code, we are unable to conduct rigorous experiments comparing DNND with PI-NDSR.
> > >
> > > **Discussion of Qualitative Comparison with DNND:** Based on your description, it seems that the idea from DNND aims to separate $F$ and $G$, generate input-output pairs, and obtain formulas from a function library using sparse regression. Therefore, DNND's approach is similar to SymDL, but SymDL utilizes genetic search while DNND employs SINDy for symbolic regression. Consequently, when the decoupled $F$ and $G$ of DNND are not accurate, its symbolic regression results will also be inaccurate. In contrast, with both the interpolated trajectories and references to decoupled $F$ and $G$ as supervision, PI-NDSR mitigates the risk of inaccurate fitting by utilizing coordinated genetic search.
> >
> > **Q6:** irregularly-sampled data.
> > > **A6:** This is indeed an advantage of our method that we did not highlight in the paper. Existing baselines, when estimating derivatives using numerical methods such as the five-point approximation, require evenly spaced sampling points. For the sake of comparison, we did not include experiments on irregularly sampled data in the previous version.
> > >
> > > We consider the SIS dynamics on a BA graph and conduct experiments using the irregular sampling method from [4]. In this scenario, our method still achieves a Rec. Prob. of 1, with an MSE of $0.901 \times 10^{-2}$, showing no significant difference compared to the result under regular sampling $(0.979 \pm 0.173) \times 10^{-2}$.
> > >
> > > [4] Neural Dynamics on Complex Networks
> >
> > **Q7:** inconsistent inputs in Eq. (2) and Eq. (4) & different meanings of G in Eq. (6) and (7).
> > > **A7:** Thanks for pointing these typos out. In Eq 4, We have modified $\phi^e \left(\mathsf{Enc}(\mathbf X(t)),t\right)$ to $\phi^e \left(\mathsf{Enc}(\mathbf x_v(t)),\mathsf{Enc}(\mathbf x_u(t)),t\right)$. In eq 6 and eq 7, we correct the typo of $G$ to $\mathsf{G}$. $\mathsf{G}$ represents the network structure, and $G$ represents one symbolic expression in the population (denoted by $\mathcal{G}$) of edge dynamics.

---

> > > ### Comment · Reviewer_FzQZ · 2024-11-26
> > > **Thanks for the response**
> > >
> > > I appreciate the author's efforts in responding and constructing new experiments on handling irregularly-sampled data and multi-dimensional network dynamics. However, I still have several major concerns that have not been clarified: (1) The design rationality for the encoder and decoder. The guarantee of $Dec(h_1 +h_2+...)=Dec(h_1)+Dec(h_2)+...$ in the paper will result in a very limited number of transformations being satisfied. For example, a linear transformation with a constant term is a counterexample, i.e., $Dec(x)=ax+b$. Therefore, I strongly doubt the necessity of decoder design. Similarly, the encoder is the inverse operation of the decoder, and it also has this problem. (2) You mentioned that “the network design part is not the main contribution of the work". However, symbolic regression is evaluated based on the output of your trained network, and different neural networks are used for comparison, making it difficult to judge whether the performance improvement is caused by symbolic regression or network design.
> > >
> > > A new question (without affecting the evaluation): Can using the Gaussian process regression in pysr to filter noise from noisy data also achieve good results?
> > >
> > > Based on the above, I think the paper still needs further revision. Although at the current stage, I think it is not ready for publication, I improve my score to encourage the author's efforts.

---

> > > > ### Author Response · Authors · 2024-11-29
> > > > **Rebuttal by Authors**
> > > >
> > > > Thanks for you response. We respond to the concerns you raised as follows:
> > > >
> > > > **Q1:** The design rationality for the encoder and decoder.
> > > >
> > > > > **A1:** Thank you for your suggestion. We do agree that this property is very important, but for the following reasons, we believe that our current design is also reasonable.
> > > > >
> > > > > **Current network satisfies additivity to some extent:**
> > > > >
> > > > >
> > > > > We think that our current network design satisfies the additivity condition to some extent.
> > > > > 1. As mentioned earlier, we only require this property to hold in the subspace in the latent space that corresponds to the input, rather than for all vectors in $\mathbb{R}^n$.
> > > > > 2. Our approach allows for some error in the estimation of node dynamics and edge dynamics, so we only require the network to have approximated rather than perfect additivity.
> > > > >
> > > > > Considering these points and our experimental results provided before, we believe that our current method satisfies the additivity condition to some extent.
> > > > >
> > > > > **Modified network does not improve performance:**
> > > > >
> > > > > We agree that enhancing the additivity of the decoder would benefit the model's performance. We have tried the following two approaches:
> > > > > 1. Adding a loss function. We tried to add a loss function to enforce the additivity of the decoder, with a weight 1 to balance the loss.
> > > > > 2. Inspired by DNND, we handle $F$ and $G$ separately with Encoder-Decoder, and instead of performing the addition in the hidden space, we add the real values obtained after decoding to combine the node dynamics and edge dynamics. This network satisfies the additivity condition strictly.
> > > > >
> > > > > We test (1) the quality of the estimated node dynamics and edge dynamics and (2) the performance of the symbolic regression. Adding a loss function did not improve the experimental results. We found that using a DNND-style network architecture provided a slight improvement for the former, but did not show significant contributions to the symbolic regression results. We attribute this to our coordinated genetic search using interpolated denoised trajectories as supervision, which allows some error in both node dynamics and edge dynamics. Therefore, we continue to use the original network design. We will explore the DNND-style network architecture more thoroughly in future work.
> > > >
> > > >
> > > > **Q2:** whether the performance improvement is caused by symbolic regression or network design
> > > >
> > > > > **A2:** Apologies for the lack of clarity in our previous response. What we meant to convey is that while the neural network component is indispensable, its structure is not the main focus of our study. Because the the proposed genetic algorithm can utilize both interpolated trajectories and node/edge dynamics reference, any neural network with good experimental performance and decoupled F and G can replace the neural network in our paper, achieving good symbolic regression results. For example, DNND may produce better results for node and edge dynamics. Therefore, the contribution from the neural network is not as significant as the genetic algorithm.
> > > >
> > > >
> > > > **Q3:** Results of using the Gaussian process regression in pysr to filter noise
> > > >
> > > > > **A3:** We experimented with the Gaussian process mentioned in PySR, applying it to smooth the trajectory of each node individually, using the same kernel function as in PySR. Our experiments showed that the Gaussian process did not achieve denoising performance comparable to the NeuralODE approach. We attribute this to the difficulty of the Gaussian process method in leveraging the trajectories of different nodes simultaneously to improve predictions. Perhaps more advanced techniques, such as Gaussian processes on graphs, could be explored in future work.
> > > >
> > > >
> > > > Thank you for your feedback. We hope our responses are helpful. If you have any further questions, please feel free to ask.

---

### Official Review · Reviewer_CABM · 2024-11-02

**Soundness:** 3
**Presentation:** 2
**Contribution:** 2
**Rating:** 6
**Confidence:** 3

**Summary:**

The paper introduces Physically Inspired Neural Dynamics Symbolic Regression, a method for automatically discovering symbolic expressions that describe complex network dynamics. The authors present a two-part solution: a Physically Inspired Neural Dynamics component that augments and denoises trajectory data through interpolation, and a coordinated genetic search algorithm that derives symbolic expressions using references from the neural dynamics. The method addresses key limitations of existing approaches by handling noisy observations from multiple trajectories without requiring pre-defined function libraries or estimated time derivatives. The authors demonstrate PI-NDSR's effectiveness through evaluation on both synthetic datasets with various dynamics and real datasets on disease spreading, showing improvements in recovery probability and error rates compared to existing methods.

**Strengths:**

- The model introduced in this paper is groundbreaking, offering a method designed to overcome limitations of other approaches, such as high sensitivity to noise.

- Comprehensive experiments are provided, comparing the proposed model with established methods across synthetic and real-world datasets.

- The paper is generally well-written (except for the conclusion) with a clear and linear structure that makes it easy to follow the proposed methodology, experiments, and results. The explanations of the Physically Inspired Neural Dynamics Symbolic Regression framework and the challenges it addresses are presented in a logical sequence, enhancing reader comprehension.

**Weaknesses:**

- The evaluation of the model is somewhat lacking, particularly in the choice of metrics for synthetic and real datasets, which is not fully explained. For example, filtering out models that don’t fully recover the skeleton in synthetic datasets may overlook valuable insights into how different methods handle structural errors. Often, a method’s ability to approximate the skeleton, even imperfectly, can still yield good performance on downstream tasks.

- The conclusion is brief, as it mainly summarizes the process without emphasizing key findings. Improvements over existing methods are not highlighted, and it lacks an impact statement or discussion of broader implications. The future work section is vague; it’s unclear what is meant by “more complex systems” and “more real-world applications.” Although limitations are mentioned, it’s not evident why the method fails for highly complex formulations.

**Questions:**

- What could be the potential issues when using the model for highly complex formulations?

- For simulated datasets, would it be better to report both filtered and unfiltered MSE, or to use a separate metric for skeleton recovery?

- It would be beneficial to expand the discussion on the limitations of the model.

---

> ### Author Response · Authors · 2024-11-25
> **Rebuttal by Authors**
>
> We thank the reviewer for valuable comments and suggestions. We respond to the concerns raised by the reviewer as follows:
>
> ### Weaknesses
> **W1:** lacking evaluation: filtered evaluation metric
> > **R1:** We choose MSE restricted to correctly recovered skeletons because the baseline methods often exhibit large MSE when recovering incorrect skeletons. Filtering out these formulas allows the baseline to achieve comparable performance and also reduces the variance of the results.
>
> **W2:** Brief conclusion, limitation, and future work
> > **R2:** We have emphasized the key findings and improvements over existing methods in the conclusion of the revision. We also added a discussion of the impact statement and detailed future work in Appendix C of the revision.
>
> ### Questions
>
> **Q1:** What could be the potential issues when using the model for highly complex formulations?
> > **A1:** The highly complex formulations indicate a large search space for the genetic algorithm. Therefore, there should be a large population size and a large number of generations for the genetic algorithm to find the symbolic expressions. Since the fitness of our method is calculated based on the pairwise combination of node and edge dynamics, the fitness evaluation is computationally expensive and memory-consuming.
>
> **Q2:** Report both filtered and unfiltered MSE.
> > **A2:** See **R1**.
>
> **Q3:** Expand the discussion of the limitations.
> > **A3:** We have added a detailed discussion of the limitations in Appendix C of the revision.

---

### Official Review · Reviewer_6eNn · 2024-11-03

**Soundness:** 2
**Presentation:** 3
**Contribution:** 2
**Rating:** 6
**Confidence:** 3

**Summary:**

This paper introduces a Physically Inspired Neural Dynamics Symbolic Regression (PI-NDSR) method for symbolic regression of complex network dynamics. The motivation behind this research is to address the challenges of symbolic regression in complex networks, particularly the issues of noisy, sparse observations and the need to decouple node dynamics from edge dynamics. PI-NDSR features two main innovations: (1) a neural dynamics module (PIND) that interpolates and denoises the observed data to generate dense, low-noise trajectories, thereby avoiding the instability of directly modeling derivatives; and (2) a coordinated genetic search algorithm that uses neural dynamics as a reference to guide symbolic regression of node and edge dynamics separately, preventing overfitting and improving search efficiency. Experiments on synthetic and real-world epidemic data demonstrate that PI-NDSR outperforms traditional methods in recovery accuracy and error rates, showing enhanced robustness in high-noise and sparse-sampling scenarios.

**Strengths:**

**Addresses a Novel Challenge of Data Sparsity and Noise in Symbolic Regression**: The PI-NDSR method effectively tackles the issue of sparse and noisy data in symbolic regression by introducing a neural dynamics module that performs interpolation and denoising, making it suitable for complex network data with inconsistent quality.

 **Improves Genetic Algorithm for Symbolic Regression**: By incorporating a coordinated genetic search that uses neural dynamics as a reference, PI-NDSR refines the traditional genetic algorithm approach, allowing for more accurate and efficient search of symbolic expressions while avoiding overfitting.

**Demonstrates Promising Performance**: The method shows competitive performance in both synthetic and real-world datasets, achieving higher recovery accuracy and lower error rates than baseline methods, indicating its potential as a robust solution for complex network dynamics modeling.

**Weaknesses:**

**Lack of Novelty in Using Neural Networks for Differential Equation Interpolation**: Modeling differential equations with neural networks to achieve interpolation is not particularly novel and has been explored in previous works. This may limit the perceived innovation in the proposed approach’s foundational methodology.

**Potential Oversmoothing from Interpolation without Fully Addressing Data Sparsity**: While the method employs interpolation to handle data sparsity, it does not adequately address how to mitigate potential oversmoothing effects in the interpolated results. This could mean that the sparsity issue remains partially unresolved, as the interpolation might overlook important variations in sparse data.

**Unclear Motivation for Decoupling Node and Edge Dynamics**: The rationale for decoupling node and edge dynamics is not clearly explained. The claim that "traditional symbolic regression methods struggle to find high-quality symbolic expressions without decoupling node and edge dynamics, leading to overfitting in symbolic space" is somewhat vague and lacks a concrete justification, potentially weakening the motivation behind this design choice.

**Questions:**

1. **Can Interpolation Truly Address Data Sparsity?**
   - The paper uses neural network-based interpolation and denoising to handle issues related to data sparsity and noise in complex network dynamics. However, interpolation fundamentally relies on existing observed data, meaning that any missing information between observation points could still lead to deviations from the true dynamics. Particularly in cases of high data sparsity, interpolation methods may produce overly smooth results, potentially missing critical dynamic variations. Thus, interpolation alone may not fully resolve the information loss caused by sparsity, as its effectiveness remains dependent on the quality and density of the original data. This could be further addressed by exploring additional complementary data sources or methods that can better capture system dynamics under sparse conditions.

2. **Unclear Motivation in the Paper’s Methodology**
   - The paper highlights challenges in symbolic regression for complex network dynamics, such as noise, data sparsity, and the need for decoupling node and edge dynamics. However, it does not fully explain the specific reasons and impacts of these challenges. In particular, the motivation for decoupling node and edge dynamics lacks sufficient detail, making it difficult for readers to understand why this decoupling is critical to obtaining high-quality symbolic expressions. Simply mentioning that "traditional methods tend to lead to overfitting in symbolic space" is somewhat vague and lacks specific examples or theoretical support. Clarifying the distinct roles of node and edge dynamics in symbolic expressions, and why decoupling is necessary for accurate representation, would enhance the paper's persuasiveness.

---

> ### Author Response · Authors · 2024-11-25
> **Rebuttal by Authors**
>
> We thank the reviewer for valuable comments and suggestions. We respond to the concerns raised by the reviewer as follows:
>
> ### Weaknesses
>
> **W1:** novelty of interpolation method
> > **R1:** The novelty of our work lies in integrating a neural network-based interpolation method with a new genetic algorithm-based symbolic regression approach to recover symbolic expressions of complex network dynamics. Our method, PI-NDSR, introduces a novel genetic search strategy that leverages both the interpolation model and references of node/edge dynamics from physically inspired neural dynamics for supervision, effectively mitigating the risk of overfitting in the symbolic space.
>
> **W2:** The sparsity issue remains partially unresolved, as the interpolation might overlook important variations in sparse data.
> > **R2:** Your concern is valid. Whether interpolation can truly address data sparsity depends on the specific data patterns. If the data has a lot of sudden changes, interpolation may not work well. In the complex network dynamics we studied, the data does not have many sudden changes, and interpolation can work well.
>
> **W3:** The rationale for decoupling node and edge dynamics
> > **R3:**
> > The rationale for decoupling node and edge dynamics is motivated by a theoretical paper [1] published in 2020 which studies the reasoning ability of DNN. This paper provides a mathematical framework explaining why certain architectures may be more suitable for particular reasoning tasks. The primary theorem (Theorem 3.6 of [1]) asserts that improved algorithmic alignment (Definition 3.3 of [1], structural correspondence between functions of a reasoning task and different modules in DNN) leads to enhanced generalization.
> >
> > Our paper wants to learn neural dynamics with strong generalization ability. Based on algorithmic alignment, Eq.(1) provides a strong inductive bias for the neural dynamics design. The designed model, Eq. (2), is algorithmically aligned with Eq. (1), allowing our model to demonstrate strong generalization. With strong generalization, the decoupling of node and edge dynamics can provide better interpolation trajectories and references for the coordinated genetic search algorithm to recover symbolic expressions of complex network dynamics. For example, the node dynamics can be seen as the dynamics on a complex network with a single node. We added this explanation to Appendix C of the revision.
> >
> > If you are interested, you can check out the blog [2] for model designs (e.g., the encoder-decoder design in our paper) and applications of algorithmic alignment.
> >
> > [1] What Can Neural Networks Reason About?
> >
> > [2] https://thegradient.pub/neural-algorithmic-reasoning
>
> ### Questions
>
> **Q1:** Can interpolation truly address data sparsity?
> > **A1:** See **R2**.
>
>
> **Q2.1:** Reasons and impacts of the challenges in symbolic regression for complex network dynamics, e.g., noise, data sparsity
> > **A2.1:** Existing methods use a five-point approximation to numerically estimate the derivative $\dot{\mathbf{X}}(t)$, which can lead to large estimation errors in the presence of noise or when the sampling is sparse. Our method, on the other hand, uses a neural network to interpolate and denoise $\mathbf{X}(t)$ instead of $\dot{\mathbf{X}}(t)$, resulting in smaller errors. An example is provided in Figure 5 of Appendix B.5.
>
> **Q2.2:** The motivation for decoupling node and edge dynamics lacks sufficient detail
> > **A2.2:** See **R3**.
>
> **Q2.2:** The example of overfitting in symbolic space
> > **A2.2:** We provide an example of overfitting in Appendix B.4.

---

### Official Review · Reviewer_1ew2 · 2024-11-03

**Soundness:** 3
**Presentation:** 3
**Contribution:** 3
**Rating:** 6
**Confidence:** 2

**Summary:**

This paper presents PI-NDSR, a novel method for symbolic regression of complex network dynamics. It leverages neural networks for trajectory interpolation and denoising, and combines it with a coordinated genetic search algorithm that utilizes references from the neural dynamics.

**Strengths:**

**Novel Approach** The use of genetic programming for symbolic regression of complex network dynamics is innovative as it constrains the symbolic search space, thereby helping to mitigate overfitting. Moreover, this approach does not rely on a predefined function library.

**Improved Performance** PI-NDSR outperforms existing methods in terms of recovery probability and error, as demonstrated by experiments on various datasets.

**Robustness** By incorporating neural ODEs, the method demonstrates strong robustness to noise and can effectively handle large time intervals between observations.

**Weaknesses:**

**Limited Novelty in Applying NeuralODE for Interpolation** The use of neural networks to model differential equations for interpolation is not particularly novel and has been addressed in prior research. With respect to the interpolation method, the paper does not introduce any methodological innovations.

**Questions:**

(1)  Have the authors attempted to explore solutions for the missing data problem?

(2) The paper mentions that the computational cost of PI-NDSR is relatively high. Could the authors provide some suggestions for reducing this computational cost?

**Details Of Ethics Concerns:**

No ethics concerns

---

> ### Author Response · Authors · 2024-11-25
> **Rebuttal by Authors**
>
> We thank the reviewer for valuable comments and suggestions. We respond to the concerns raised by the reviewer as follows:
>
> ### Weaknesses
>
> **W1:** novelty of interpolation model
> > **R1:** The novelty of our work lies in integrating a neural network-based interpolation method with a new genetic algorithm-based symbolic regression approach to recover symbolic expressions of complex network dynamics. Our method, PI-NDSR, introduces a novel genetic search strategy that leverages both the interpolation model and references of node/edge dynamics from physically inspired neural dynamics for supervision, effectively mitigating the risk of overfitting in the symbolic space.
>
> ### Questions
>
> **Q1:** Have the authors attempted to explore solutions for the missing data problem?
> > **A1:** Current methods are unable to handle the missing data problem. In future work, we will explore potential solutions such as introducing latent variables into the network dynamics.
>
> **Q2:** Suggestions for reducing this computational cost
> > **A2:** The computational cost of PI-NDSR is mainly from two parts, i.e., neural dynamic training and symbolic regression with genetic search. In the future, we will explore improving the computational efficiency of genetic search by incorporating more advanced symbolic regression methods, e.g., transformer-based and LLM-based symbolic regression methods.

---

### Official Review · Reviewer_m6m5 · 2024-11-04

**Soundness:** 3
**Presentation:** 2
**Contribution:** 2
**Rating:** 6
**Confidence:** 3

**Summary:**

The paper introduces Physically Inspired Neural Dynamics Symbolic Regression (PI-NDSR), a new method for symbolic regression of complex network dynamics. The approach has two main components: neural dynamics, which denoises and augments observed data, and a coordinated genetic search, which leverages the previously build neural references to optimize the search for symbolic expressions. Evaluated on synthetic an real-world datasets, PI-NDSR outperforms previous methods, offering more accurate symbolic representations and robustness against noise and sampling irregularities.

**Strengths:**

The results on these datasets show the good performance of the method, with some proof of robustness too. This applies to the real world dataset too, where the extracted dynamics of this method don't have "un-physical" properties. The ablation studies show that both parts of the algorithm are necessary for the best performance

**Weaknesses:**

It is not clear if this method could scale to real world datasets that have more complex dynamics, since there is only a single real-world dataset.

**Questions:**

Did I understand correctly that to train PIND you use four networks? An Encoder, a Decoder, $\phi^e$, $\phi^n$? I believe section 4.1 could be slightly expanded to show how the neural dynamics is trained, maybe supported with a figure.

I also found these typos:
In section 5.1, in "baseline": metods should be methods. "can not be aplied directly" should be "cannot be applied directly".
In Appendix A2, "NerualODE" should be "NeuralODE"

---

> ### Author Response · Authors · 2024-11-25
> **Rebuttal by Authors**
>
> We thank the reviewer for valuable comments and suggestions. We respond to the concerns raised by the reviewer as follows:
>
> ### Weaknesses
>
> **W1:** It is not clear if this method could scale to real-world datasets that have more complex dynamics, since there is only a single real-world dataset.
> > **R2:** Thank you for your suggestion. Our approach can indeed be applied to more real-world datasets, aiding scientists in understanding complex real-world systems. However, this often requires extensive data collection efforts. We are collaborating with experts in related fields to explore the application of our method to brain networks, transportation networks, and protein network data. Due to the inherent complexity of these tasks, we consider them part of our future work.
>
> ### Questions
>
> **Q1:** details about about training 4 networks
> > **A1:** We combine the four networks into a single network, which is trained end-to-end with $\min_{\theta}\sum_{v\in V, t\in \mathcal{T}}\|f_\theta (\mathsf{G}, \mathbf X(t_0), t)_v - x_v(t)\|_1$. We have added details to how the neural dynamics are trained in Section 4.1 of the revision.
>
> **Q2:** typos in the paper
> > **A2:** Thank you for pointing that out. We have corrected this in the revised version.

---

> > ### Comment · Reviewer_m6m5 · 2024-11-27
> >
> > I appreciate the Authors comments and understand that gathering real world datasets is out of the scope for this paper, but could be an interesting future work. I also appreciate the corrections made. I stand by my previous overall score

---

### Comment · Area_Chair_cfyW · 2024-11-25

Dear reviewers,

A reminder that **November, 26** is the last day to interact with the authors, before the private discussion with the area chairs. At the very least, please acknowledge having read the rebuttal (if present). If the rebuttal was satisfying, please improve your score accordingly. Finally, if you have concerns that might be solved in time, this is the last chance before moving on to the next phase.

Thanks,
The AC

---

### Meta-Review · Area_Chair_cfyW · 2024-12-14

**Metareview:**

The paper proposes a framework for symbolic regression of network dynamics using graph neural networks and genetic algorithms (for extracting a symbolic representation).

The paper had a large amount of reviews (6), which scores ranging from 3 to 6. After the rebuttal phase, the most negative score improved to 5, with 2 reviewer negative on the current status of the paper. They both raise a few concerns that should be addressed. In general, there was limited interaction in the rebuttal stage, and as a result some concerns remained open.

There is a consensus that the paper's novelty is unclear, and that the design of the method requires more details (especially concerning the assumptions that are made). The authors have commented that the use of a genetic algorithm conditioned on the specific neural formulation is novel enough. All reviewers who have interacted during the rebuttal have remained mostly negative.

Due to the unaddressed concerns and the generally unsatisfactory rebuttal, I have leaned towards a negative evaluation of the paper, although I mention this is a borderline case which could have benefitted from a stronger discussion.

**Additional Comments On Reviewer Discussion:**

- **Reviewer KBLS** is concerned about novelty and insufficient experimental evaluation. The authors gave a quick rebuttal which had no effect on the score. *I partially agree on the novelty* concern and this impacted my final decision.

- **Reviewer FzQZ** is mostly concerned about the design of the decoder (which requires a linearity assumption in the latent space of the model) and a lack of comparison with some related works. Also in this case, the rebuttal was considered only partially insufficient, although there was limited discussion. I don't have a strong opinion in either direction, so *I trust the reviewer* in this case.

- **Reviewer CABM** was mostly concerned about the choice of metrics and was not involved in the rebuttal. I found the review not very convincing and *it did not influence my final evaluation* (although it's still in line with the other reviewers).

- **Reviewer 1ew2** made a very short review lamenting a lack of novelty and was not involved in the rebuttal. Due to the low quality of the review, *I ignored the review in the final decision*.

- **Reviewer 6eNn** had three major concerns on novelty, possible overfitting, and on the design of the method. There was no interaction in the rebuttal, so most of these concerns remained open.

- **Reviewer m6m5** was mostly concerned about a "lack of real-world datasets". I think the rebuttal here was convincing (as the benchmarks are in line with existing literature), so *this review had limited impact* on my final evaluation.

---

### Decision · Program_Chairs · 2025-01-22

Reject